# FOURIER NETWORKS FOR UNCERTAINTY ESTIMATES AND OUT-OF-DISTRIBUTION DETECTION

## ABSTRACT

A simple method for obtaining uncertainty estimates for Neural Network classifiers (e.g. for out-of-distribution detection) is to use an ensemble of independently trained networks and average the softmax outputs. While this method works, its results are still very far from human performance on standard data sets. We investigate how this method works and observe three fundamental limitations: "Unreasonable" extrapolation, "unreasonable" agreement between the networks in an ensemble, and the filtering out of features that distinguish the training distribution from some out–of–distribution inputs, but do not contribute to the classification. To mitigate these problems we suggest "large" initializations in the first layers and changing the activation function to $\sin(x)$ in the last hidden layer. We show that this combines the out-of-distribution behavior from nearest neighbor methods with the generalization capabilities of neural networks, and achieves greatly improved out-of- distribution detection on standard data sets (MNIST/fashionMNIST/notMNIST, SVHN/CIFAR10).

## 1 INTRODUCTION

When a neural network classifies inputs, we often need to have some measure of the uncertainty involved in the prediction. In particular, in safety critical applications we would like to know when the classifier received an input that is different from the inputs it has been trained on - in such cases it could be dangerous to just use the "best guess", instead we may want to choose some safe action or involve a human.

One solution for this "out-of-distribution detection" problem could be to add a label "unknown" and augment the training set with inputs that do not belong to any of the labels of the classifier and which get the new label "unknown". However, the danger is that the classifier learns the particular type of "known unknowns" it was trained on and does not generalize to the "unknown unknowns". To avoid this, we treat out-of-distribution detection as a "one class classification" problem, i.e. we want to learn the input distribution without help of out of distribution inputs.

The simplest approach is to use the softmax output of a neural net as a confidence measure. It is well known that the softmax output tends to be "overconfident", so we cannot interpret it directly as a probability for the chosen class. However, it still tends to be more confident for samples from the correct distribution than for outliers, so setting a threshold for the softmax output can to some extent distinguish between "in distribution" and "out-of-distribution" samples. This is the "baseline method" for outlier detection suggested in (Hendrycks & Gimpel, 2017).

In (Lakshminarayanan et al., 2017) this baseline is improved by using an ensemble of classifiers (and adversarial training): Often each random initialization of a neural network gives an overconfident classifier, but different classifiers disagree - averaging the softmax output over an ensemble of classifiers then gives an improved signal.

Other, more computationally demanding approaches are Bayesian Neural Networks ((Neal, 1996), (Barber & Bishop, 1998), (Blundell et al., 2015)), nearest neighbor methods (e.g. (Mandelbaum & Weinshall, 2017), (Jiang et al., 2018), (Papernot & McDaniel, 2018), (Frosst et al., 2019)). For some further (less closely) related works, see appendix A.

We investigate limitations of the "ensemble averaged softmax output" method, as this seems to hit the sweet spot of being easy and fast, but still giving good results. To overcome the observed limitations and further improve this method, we propose two changes:

- Changing the activation function of the last hidden layer to the $\sin(x)$ function.
  (We will see in the next section how this can guard against overgeneralization.)
- Use larger than usual initialization, to increase the chances of obtaining more diverse networks for an ensemble.

In the next three sections we go through three observed problems with ensembles of ReLU networks and explain how these changes mitigate them, and in the following sections we evaluate the suggested changes on standard data sets (MNIST, CIFAR10, etc.).

We evaluate this uncertainty mainly with view to out-of-distribution detection: If we have meaningful uncertainties, the classifier should in particular "know when it does not know the answer". This means it should display uncertainty for inputs that do not "belong to the same type of input" as the training examples. By selecting different thresholds for the uncertainty, we can plot the ROC curve by recording for each threshold the fraction of samples below the threshold for "in distribution" ($x$) and for "out-of-distribution" ($y$) inputs. In particular, figure 1 gives this evaluation for our method against the standard ensemble methods on two standard data sets.

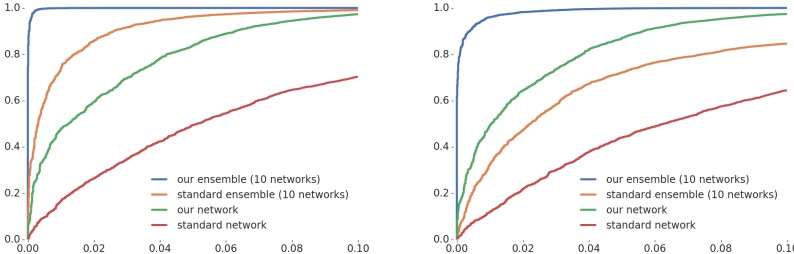

Figure 1: ROC curves for networks ensembles trained on MNIST,
evaluated on notMNIST (left) and fashionMNIST (right).
$x$-axis: MNIST samples recognized as not belonging to MNIST,
$y$-axis: notMNIST samples recognized as not belonging to MNIST.

## 2 PROBLEM: "UNREASONABLE" EXTRAPOLATION

We first look at the simplest possible example of classification: The training set consists only of one point $x_0 \in \mathbb{R}^d$ which has label "1" (we assume there is another label, for which we do not have a sample). Training standard ReLU networks with one hidden layer on just one point $x_0 = 2 \in \mathbb{R}$ we get something like figure 1: Gradient descent finds a simple functions describing the data, which will (with very high probability) be an increasing function of $x$. So for larger values of $x$ the network will become arbitrarily certain that the label of $x$ should also be "1". (See appendix C for more details.)

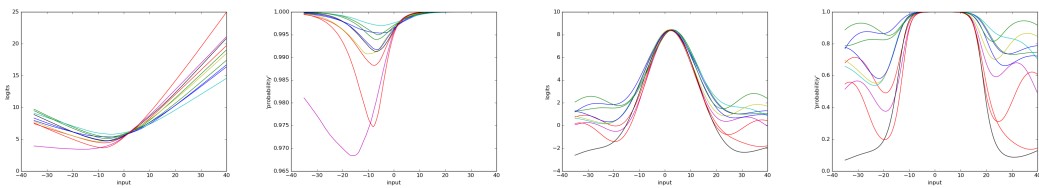

Figure 2: Logits and probabilities from 10 randomly initialized ReLU–networks (left) and Fourier–networks (right). $x$-axis: Input $x$, $y$-axis: Estimated logits/probabilites for input $x$.

It has been observed in (Hein et al., 2018) that this is true in general: Given any ReLU network trained on some data for a classification problem in $\mathbb{R}^d$, it will describe a piecewise linear function. If we look at the values of this function along a ray to infinity, this will be a linear function when we are far enough from the origin, and in general this means the network will become arbitrarily certain

of its classification for all points on the ray far enough from the origin. This is clearly not desirable, we would rather get a low confidence when we are far from the training data. (Here "far from the training data" is a convenient aspect that can easily be checked mathematically. Since the data are often bounded, it is actually more relevant in practice that this behavior starts as we leave the range of the training data. As an illustration, see figures 3 and 4.)

Going back to our simplest possible special case, what should our confidence be that other points belong to the same label? Of course, there is no unique "correct" answer, but a reasonable model could be that our confidence decays like a Gauss function around this point $\vec{x}_0$, i.e. is proportional to

$$e^{-|\vec{x}-\vec{x}_0|^2/2\sigma^2} \tag{1}$$

for some $\sigma$ that defines the scale on which our confidence decays.

One way to force such a decay would be to use directly Gauss functions (equation 1) as activation functions, which leads to RBF networks which we could consider as a form of nearest neighbor methods. While they recognize reliably points that are not close to one of the $\vec{x}_0$ in (equation 1), they do not generalize as well as ReLU networks (see appendix G), so we will use a more indirect approach.

Using that the Fourier transform of such a Gauss function is again of this shape, we can write equation 1 also as an expectation:

$$e^{-|\vec{x}-\vec{x}_0|^2/2\sigma^2} = \mathbb{E}_{\vec{w}\sim\mathcal{N}(0,\Sigma)}\Big[\cos\big(\vec{w}\cdot(\vec{x}-\vec{x}_0)\big)\Big],$$

where $\Sigma$ is the diagonal covariance matrix $\sigma^{-2}\cdot I$. When we approximate the expectation by a finite sum over $N$ samples $\vec{w}_i$ from $\mathcal{N}(0,\Sigma)$ we get an expression of the form

$$f(x) = \sum_{i=1}^{N} u_i \cdot \cos(\vec{w}_i\cdot\vec{x}+b_i)$$

where $u_i = 1/N$ and the $b_i$ are chosen such that the $\cos(\vec{w}_i\cdot\vec{x}_0+b_i) = 1$. This function can be interpreted as the output of a neural net with one hidden layer of $N$ neurons with activation function $\cos(x)$ (we will later shift the phase $b_i$ and use $\sin(x)$). (The output is one dimensional since we only care about the logits for the one label that appears in our training set.) This is the main idea in the proof of the following result (see appendix C for details and proof):

**Proposition 1**: Assume the weight vectors $\vec{w}_i$ are sampled from a normal distribution $\mathcal{N}(0,\Sigma)$ and then fixed, the numbers $b_i$ and $u_i$ are sampled independently from some distribution (not identically zero) with finite second moments (e.g. a normal distribution) and then trained on one data point $\vec{x}_0$ with label 1, then Gradient Descent (for the usual cross-entropy loss) will make the network's output converge to a function that is approximately proportional to $e^{-\vec{x}^T\Sigma\vec{x}/2}$. This approximation becomes exact when the number of neurons approaches infinity, or when we take the expected value. ∎

We also have for general training sets (see appendix C for a proof):

**Proposition 2**: Under the same idealized conditions (infinite number of neurons, weights $\vec{w}_i$ frozen), but arbitrary finite training set, and assuming the training converged, the function goes to 0 for $|\vec{x}| \to \infty$. ∎

To get an exact formula, we froze the weights $\vec{w}_i$. In practice we would train them as well, but at least experimentally we do not see a difference when the $\vec{w}_i$ are large: They do not get changed by a large amount, and the result still looks the same. An example is given in the two plots of the right side of figure 2. When we train on a training set of a few points, we see experimentally that (under the same idealized conditions: Large number of hidden neurons, frozen $\vec{w}_i$) we get an approximation to a weighted sum of Gaussian functions at these points (see appendix E). So it may seem we constructed a complicated approximation to a nearest neighbor classifier.

We now switch the activation function to $\sin(x)$, this makes no difference for the function if we compensate by shifting the $b_i$ by $\pi/2$, but the region of "small initialization" $|\vec{w}_i|, b_i \approx 0$ is now similar to a linear network: The input to the hidden neurons $i$ will be $\vec{w}_i \cdot \vec{x} \approx 0$, and $\sin(\vec{w}_i \cdot \vec{x}) \approx \vec{w}_i \cdot \vec{x}$. As training progresses, $|x|$ usually increases. For ReLU activation, this increase is not bounded (if we are on the side $\vec{w}_i \cdot \vec{x} > 0$), but for $\sin$ the increase will usually stop at the first

maximum or minimum of $\sin$, i.e. around $\approx \pm\pi/2$ (or also at a different minimum/maximum if we start from higher initialization, see appendix P for experimental examples). This means that on the training set, features (i.e. the output of the $\sin(x)$ neurons in the last hidden layer) get a large value by tuning the input to around $x \approx \pm\pi/2$, which is difficult to achieve "accidentally" for input that is not in the training set. By contrast, for ReLU (or tanh) activation, the inputs only have to achieve any high value.

We will call networks in which the last hidden layer uses the $\sin(x)$ activation function "Fourier networks".

So the Fourier network is somewhere between a ReLU network (small initialization) and a nearest neighbor classifier (large initialization). In particular for intermediate initializations it generalizes better than a nearest neighbor classifier, but still detects more outliers than a ReLU network, which leads to an out-of-distribution detection which is better than either of them, as we will see later (see in particular appendix G for a comparison to nearest neighbor classifiers).

## 3   PROBLEM: "UNREASONABLE" SIMILARITY OF NETWORKS IN THE ENSEMBLE

In section 2 we considered only one network, and saw that ReLU networks become arbitrarily certain far from the training set. However, averaging the softmax output over an ensemble of networks could fix that: Even if each network is very certain about its prediction, the ensemble could give low confidence if the different networks disagree. This indeed happens, but less often than one might expect. To illustrate this, we generate the input in figure 3, which has 20 clusters of points with labels "red", "green", or "blue".

Input

Ensemble of ReLU Networks

Ensemble of Fourier Networks

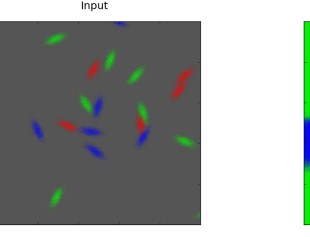 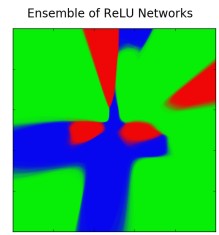 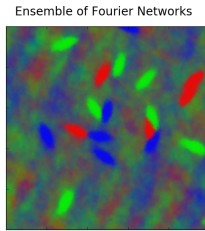

Figure 3: A 2-dim classification problem. Left image: Input, color indicates label.
Following images: Average output of an ensemble of 50 networks.

We take an ensemble of 50 networks that were independently trained on the same data set, and use the average "probabilities" for the 3 possible labels as RGB values for the corresponding pixels. In the second image (ReLU networks) the ensemble is still overconfident in most of the area, whereas on the right the ensemble of Fourier networks is only confident close to the input.

The reason for the overconfidence of the ReLU ensemble is that the individual networks tend to agree even far from the sample points. On the other hand, the Fourier networks find different sparse Fourier interpolations to describe the input, so their output only agrees close to the inputs, see figure 4. For another 2-dimensional example, see also appendix N.

As a higher dimensional illustration, we train 50 ReLU networks with standard initializations on MNIST images with labels from 0 to 4. Evaluated on MNIST images for labels 5 to 9, for a quarter of all images all 50 classifiers agree on the same label (which is of course wrong, since it is one of known labels $0-4$). If we use smaller initialization, all 50 classifiers agree even for the majority of inputs.

We can understand why this similarity of independently initialized networks is particularly prevalent for "small" initializations: We expect that Gradient Descent finds local optima "close to" the initial point. If we start with high random initializations, we may have a better chance that they are far apart and converge to different solutions, but for small initialization the networks are likely to converge to

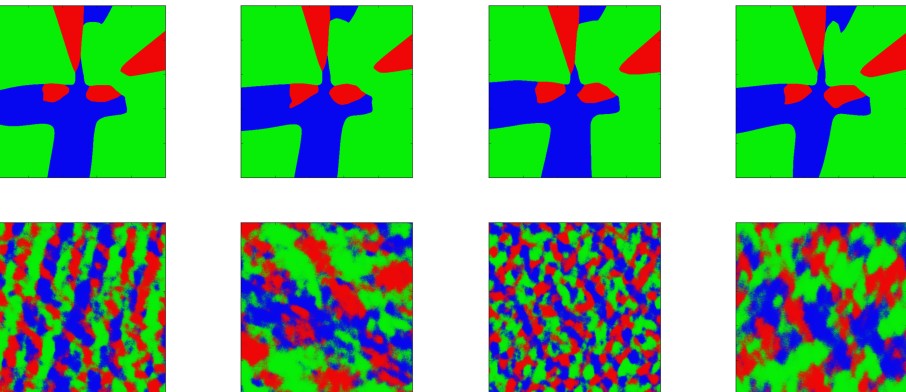

Figure 4: Predictions of 4 randomly initialized networks.
Upper row: ReLU networks, lower row: Fourier networks.

the same closest solution. This can be made precise for ReLU networks with one hidden layer and "infinitesimal" initialization, see appendix H.

As a method between RBF networks and Fourier networks, we can also try using a Gauss function as an activation function. While this does show some of the benefits of Fourier networks, extrapolating with Gauss functions predictably reverts to zero, whereas the sum of periodic functions become essentially random far from the training distribution, resulting in far more diverse ensembles, see appendix F, figure 10.

## 4 PROBLEM: CONSTANT FUNCTIONS ON THE TRAINING MANIFOLD

The third problem we see may be a general problem for using networks trained as classifiers also for out–of–distribution detection: If we have a feature (function) that is constant on the training distribution, but takes on other values out–of–distribution, we can use it to detect out–of–distribution inputs. However, when we train for distinguishing between different parts of the input distribution, such a feature is useless and may be dropped during training.

How do such features arise? One simple possibility is that some part of the input is constant. For example, for MNIST images, the corner pixels are always white. But in notMNIST or fashionM-NIST this is not always true (see e.g. the "E" in figure 17), and thus this gives an easy way to detect images not in MNIST.

Another possibility is that there may be different ways to come to the conclusion that an input gets a certain label $L$. For example, for MNIST we usually can already guess the input has label "0" when we only see the top or the bottom half of the image. Let us say features $f$ and $g$ get value 1 if their clues to detect label $L$ is present in the image, and are 0 otherwise. Then $f - g$ is always 0 on the training set (as they are 1 if and only if the image is of label $L$), but it may take on different values out–of–distribution. (For some preliminary experiments with forcing such different features into separate networks, see appendix O).

Why would such features be "dropped"? If we start from small initialization, only weights that are useful for distinguishing between labels will be increased. So e.g. if the corners are always white, there is no incentive to change the weight that connects a corner pixel to the next layers, and if we start from small initialization, these pixels will never contribute to the end result.

So an obvious mitigation is to use larger initialization: Although the weights will still not be changed, the pixel will likely have an influence on the end result, and in different networks of the ensemble this influence will be different. So if the pixel is not white, this may lead to the different classifiers disagreeing about the label.

## 5 EXPERIMENTS: TWO HIDDEN LAYERS, FULLY CONNECTED

We follow the example given in (Lakshminarayanan et al., 2017) and look at fully connected networks with two hidden layers of 200 neurons, which we train on MNIST. For the baseline network, we use the model presented in (Lakshminarayanan et al., 2017), namely ensembles of networks with ReLU activations and the standard initialization of the weights ($\sigma = \sqrt{2/n}$ with $n = 784$ for the first layer, $n = 200$ for the second layer). For the Fourier network we change the activation function of the deeper layer to $\sin$ and use $\sigma_1 = 0.75$ and $\sigma_2 = 0.0002$.

We train the networks on the training set of MNIST (60000 images), and evaluate them both on the test set of MNIST and on other data sets, here we use notMNIST (see (Bulatov, 2011)) and fashionMNIST (see (Xiao et al., 2017)) as the "outlier" data sets. For other data sets see appendix K). The ROC curves that are obtained this way are presented in figure 1. The variation of these curves is captured by figure 15 in appendix I.

While using an ensemble of ReLU nets gives a very substantial improvement over using one ReLU net, the result is still not close to what humans would achieve: The classifiers used here have some 98% accuracy, if we allow not recognizing an image as MNIST digit in 2% of the cases, we might expect to recognize that the printed letters of the alphabet are no handwritten digits in most cases, but in the above graph 2% error rate for MNIST corresponds to "only" 80% success rate on notMNIST. However, the ensembles of Fourier networks recognizes almost all of notMNIST images if we allow 2% error rate on MNIST.

In appendix M, figure 22 shows the corresponding results for training on the more challenging data set fashionMNIST, the results are qualitatively the same.

In the literature, the performance of such "out-of-distribution" detection is often measured by the area under the ROC curve, in Table 1 are numbers for MNIST vs. notMNIST.

| Paper | Method | ID set | OOD set | AUROC |
|-------|--------|--------|---------|-------|
| (Hendrycks & Gimpel, 2017) | baseline | MNIST | notMNIST | 93.2% |
| (Liang et al., 2018) | outlier exposure | MNIST | notMNIST | 98.2% |
| ours | Fourier networks | MNIST | notMNIST | **99.9%** |
| (Malinin & Gales, 2018) | outlier exposure | MNIST | FashionMNIST 5-9 | 96.5% |
| ours | Fourier networks | MNIST | FashionMNIST 5-9 | **99.7%** |

Table 1: Area under ROC curve for entropy as the OOD detection score.

Some of the approaches in Table 1 use outliers for training/tuning the model, so they are dependant on how similar the outliers used for training are to those used at test time. For more details about the baselines we refer the reader to Appendix R.

## 6 EXPERIMENTS: EFFECT OF INITIALIZATION VS. ACTIVATION

If we use only one hidden layer, we can only vary the $\sigma$ for weights between the input and the hidden layer (we keep the initialization of the output layer at the standard $\sigma = \sqrt{2/n}$). This makes it easy to plot how much of the effect is due to "high initialization" vs. the choice of activation function.

We plot the area under the ROC curve in dependence on initialization: On the $x$-axis we plot $\log(\sigma)$, on the $y$-axis the negative logarithm of the "error" := area above the ROC curve.

The error bars correspond to two standard deviations (we has used randomly sampled ensembles of 10 networks out of 50 pre-computed networks).

We see in the first three images of figure 5 that notMNIST images are the easiest to detect, and flipped digits from MNIST are the most difficult, but in each case the Fourier networks clearly outperform the ReLU networks. One reason why notMNIST is particularly easy probably is that its images often can be recognized by just looking at pixels close to the corners: In MNIST these pixels are always white, but in notMNIST often not. To eliminate this shortcut, we also consider "masked" versions of notMNIST and fashionMNIST in which these pixels are set to white (see appendix K for details). The rightmost image in figure 5 shows the corresponding graphs for the

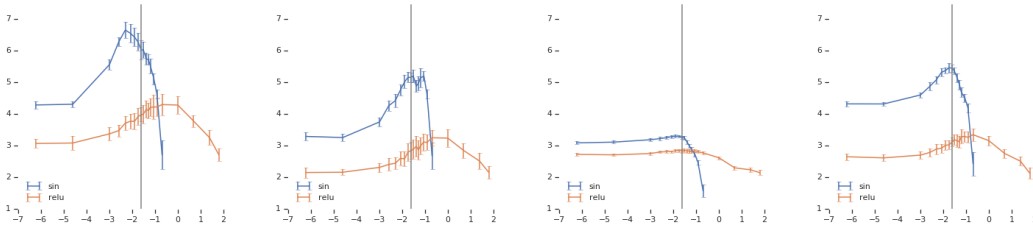

Figure 5: Influence of initialization and activation function
$x$-axis: $\log(\sigma)$, $y$-axis: $-\log(\text{Area above ROC curve})$, higher is better
Left to right: notMNIST, fashion MNIST, flipped MNIST, masked notMNIST

masked version of notMNIST, see figures 20 and 21 in appendix L for masked fashionMNIST and images in which each pixel is independently sampled from MNIST images, so each pixel has the same value distribution as in MNIST. Even with the "looking at the corners shortcut" disabled, the Fourier networks still perform well.

The vertical line corresponds to $\sigma = 0.2$, which seems to be close to optimal for Fourier networks for all test sets. This corresponds to a standard deviation of around 1.4 for the input to the $\sin(x)$ activation function (for our encoding of the MNIST pixels, which we scale into the interval $[-1, 1]$).

We also see that "larger than usual" initialization alone already helps for ReLU networks, but using the $\sin(x)$ activation function gives a significant additional benefit. (See also appendix N for more general initialization schemes.)

## 7 EXPERIMENTS: CONVOLUTIONAL NETWORKS ON SVHN VS CIFAR10

The same phenomena described in the previous section for fully-connected networks apply for convolutional networks as well, thus allowing one to reap the benefits of using Fourier networks for out-of-distribution detection on more complex image datasets. To illustrate this, we train ensembles of classifiers on the Street View House Numbers dataset (SVHN) (Netzer et al., 2011) and use CIFAR10 (Krizhevsky et al., 2009) as out-of-distribution samples. Once again, we compare against a baseline of ReLU networks, first described as performing well on this task in (Lakshminarayanan et al., 2017).

Three different convolutional architectures have been considered, with increasing complexity: i) a 2-layer convolutional block followed by a fully-connected layer with ReLU activations and one with Sine activations dubbed *SimpleCNN*, ii) a network containing the first 4 layers from the classic VGG16 architecture (Simonyan & Zisserman, 2015) followed by a sine fully-connected layer referred to as *ShallowVGG*, and iii) the full-fledged VGG16 network. Appendix J contains the exact hyperparameter values used for each network.

Table 2 presents the best results obtained using this setup for both the ReLU and Fourier networks. Out-of-distribution samples are detected using two approaches, by selecting a threshold for either the maximum probability as obtained after aggregating the outputs of the softmax layers in an ensemble (presented in appendix Q) or for the entropy of the ensemble's prediction (presented in Table 2). For evaluation, we used the area under the ROC curve (*AUROC↑*) as well as the false positive rate at 80% true positive rate (*FPR80↓*). The arrows indicate how the metrics change as the model gets better (i.e. ↑ if a larger value is better and ↓ if a smaller value is better). Additional results are presented in appendix Q.
One other way in which one can assess qualitatively the performance of an out-of-distribution detection approach is by inspecting the distribution of the entropy of the softmax outputs over in-distribution and out-of-distribution samples. Ideally, the entropy for in-distribution samples stays close to 0 (meaning that the model is certain about its predictions) while the one for out-of-distribution samples is pushed as far from 0 as possible (the maximum entropy that a 10-class classifier can achieve is $\log 10 \approx 2.3$).

Figure 6 shows how Fourier networks compare to regular ReLU networks for the Shallow VGG architectures (with $\sigma = 0.01$ used as the standard deviation of the initialization distribution of the last layer). The Fourier networks make lower certainty predictions about out-of-distribution samples.

| Network | Activation | AUROC↑ | FPR80↓ |
|---|---|---|---|
| Simple CNN | ReLU | 93.2% | 10.3% |
| | $sin(x)$ | **96.6%** | **5.17%** |
| Shallow VGG | ReLU | 93.5% | 9.97% |
| | $sin(x)$ | **95.8%** | **6.71%** |
| VGG16 | ReLU | 95.7% | 6.73% |
| | $sin(x)$ | **95.9%** | **6.51%** |

Table 2: Out-of-distribution performance on CIFAR10, for ensembles of 5 models trained on SVHN.

The histograms for the ReLU networks are similar to the ones reported by (Lakshminarayanan et al., 2017) on the same task using ensembles of networks, with or without adversarial training. The curves correspond to ensembles of 1, 5, and 10 models.

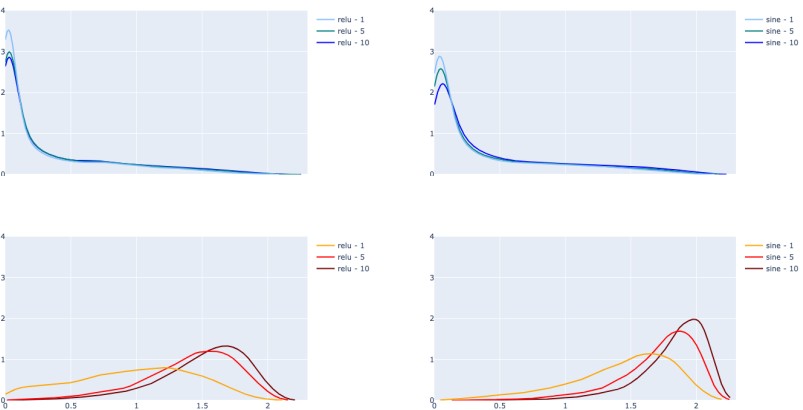

Figure 6: Entropy histograms for SVHN as in-distribution set (top row) and CIFAR10 as out-of-distribution set (bottom row). Comparison between using $sin(x)$ (right-hand side) or ReLU activations (left-hand side) on the last layer of a Shallow VGG network.

## 8 CONCLUSION

We suggested using the $sin(x)$ activation function in the last layer, and larger than usual weight initializations to mitigate three problems we saw in using ensembles of ReLU networks for out–of–distribution detection:

- "Unreasonable" extrapolation:
  While ReLU networks tend to get more confident away from the training set, Fourier networks get decreasing confidence in the average.
- "Unreasonable" agreement between the networks in an ensemble:
  Larger initialization can make the networks more diverse, and the mixture of $sin(x)$ functions with different frequencies makes the networks more random away from the training set.
- Filtering out of features that distinguish the training distribution from some out–of–distribution inputs, but do not contribute to the classification:
  Larger initialization can preserve such features, they do not influence the in–distribution classification, but create diversity in the ensemble when applied to out–of–distribution input.

We showed that this combines the out-of-distribution behavior from nearest neighbor methods with the generalization capabilities of neural networks, and achieves greatly improved out-of-distribution detection on standard data sets (MNIST/fashionMNIST/notMNIST, SVHN/CIFAR10).

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

## A  APPENDIX A: MORE RELATED WORKS

Related to (Lakshminarayanan et al., 2017) is the suggestion of "MC dropout" (Gal & Ghahramani, 2016), here the ensemble of networks is replaced by one network, from which different predictions are produced by Dropout at prediction time. This is faster in training and evaluation, but in general not as effective as training the networks independently.

Another method that is also suggested in (Lakshminarayanan et al., 2017) is adversarial training, but its effect seems to be small compared to the effect of using ensembles instead of single networks.

Using a RBF layer on top of a normal network was first suggested in (Y. LeCun & Haffner, 1998), and is further investigated in (Zadeh & Hosseini, 2018).

While we are avoiding the use of "outliers" as training input, it can be helpful. A recent work in this direction is (Hendrycks et al., 2019). "Interesting outliers" can also be generated by a GAN, see (Lee et al., 2018) and (Kliger & Fleishman, 2018).

We focused on the use of uncertainties for out–of–distribution detection, and for this purpose it does not matter whether the "probabilities" are really correct (as long as they are higher for in–distribution than for out–of–distribution). However, if one wants "correctly scaled" probabilities, one can try to learn the right scale from the training data, see (Pawlowski et al., 2018).

We used the classifying networks and their "probability" output also for one–class–classification, as this is the computationally cheapest solution. But another intuitive approach is to use a generative model to produce a probability that an input belongs to the training distribution (Bishop, 1994). However, (Choi et al., 2018) and (Nalisnick et al., 2018) showed how this can fail and actually produce higher probabilities for out of distribution inputs. An interesting method that fixes this problem was recently described in (Ren et al., 2019).

For more works related to $\sin(x)$ as activation function, see the next section.

## B  ACCURACY AND TRAINING WITH SIN AS ACTIVATION FUNCTION

We did not investigate in detail the general behavior of the $\sin(x)$–function as an activation function with respect to the usual accuracy and training behavior, but in our experiments, we did not see any significant difference to $ReLU(x)$ networks inside the training distribution.

For example, the table below shows the classification accuracy of ensembles of 5 ReLU networks and Fourier networks on the SVHN test set. For each case, the accuracy is reported for the ensemble with the best out-of-distribution detection performance (the same that are presented in Table 2).

| Network | ReLU network | ReLU ensemble | Fourier network | Fourier network ensemble |
|---------|--------------|---------------|-----------------|--------------------------|
| Simple CNN | 89.38% | 91.9% | 89.50% | 91.9% |
| Shallow VGG | 89.10% | 90.7% | 89.60% | 90.8% |
| VGG16 | 86.21% | 90.6% | 86.47% | 90.5% |

Table 3: Classification accuracy of individual networks (averaged over 5 instances) and ensembles of 5 models on SVHN.

These observations are consistent with the remarks in chapter 6.3.3. "Other Hidden Units" in (Goodfellow et al., 2016).

It is usually said that $ReLU$ works much better than other activation functions for deep neural nets, but that is less relevant for us since we only use $\sin(x)$ in the last layer.

Nevertheless, there may be special cases in which $\sin(x)$ behaves better or worse than ReLU, as has been observed in e.g. (Sopena et al., 1999) and (Parascandolo et al., 2017).

Another way to formulate what a $\sin(x)$–layer does, is that it computes a sparse Fourier approximation. Sparse Fourier approximation has been extensively studied (e.g. (Gilbert et al., 2002), (Candes et al., 2006)), but in a quite different context: Usually the aim is to fully reconstruct a function from a small amount of input / output data. This is possible when the function is known to have a sparse

Fourier representation. In our context it is unlikely that our functions have a sparse Fourier representation, and instead of trying to reconstruct the function the aim is rather to use different sparse Fourier interpolations to show the uncertainty we have in areas where we do not have data.

## C  Proofs of Proposition 1 and 2

We study the Gradient Descent dynamic of a neural net with

- input $\vec{x} \in \mathbb{R}^d$
- one hidden layer with $N$ neurons with activation function $\cos$,
  connected to the input layer by fixed weights $\vec{w}_i \in \mathbb{R}^d$, $i = 1, 2, ..., N$
- one output $f(\vec{x}) \in \mathbb{R}$, connected to the hidden layer with weights $u_i \in \mathbb{R}$.

The output of this neural net at the point $\vec{x}$ is given by

$$f(\vec{x}) = \sum_{i=1}^{N} u_i \cdot \cos(\vec{w}_i \cdot \vec{x} + b_i)$$

(We are using only one output dimension since for Proposition 1 we will be training it only on one data point, so the outputs for other labels do not matter for now.)

We use the cross entropy loss, which is for a data set of one point $\vec{x}_0$ with the correct label given by

$$L = \log \left( 1 + e^{-f(\vec{x}_0)} \right).$$

The derivative of $L$ for some parameter $\theta$ (i.e. one of the $b_i$ or $u_i$) is

$$\frac{\partial}{\partial \theta} L = -\frac{1}{1 + e^{f(\vec{x}_0)}} \cdot \frac{\partial}{\partial \theta} f(\vec{x}_0).$$

With $d_i := \vec{w}_i \cdot \vec{x}_0$ the last derivative is

$$\frac{\partial}{\partial b_i} f(\vec{x}_0) = -u_i \cdot \sin(d_i + b_i)$$

$$\frac{\partial}{\partial u_i} f(\vec{x}_0) = \cos(d_i + b_i)$$

We will assume "infinitesimal" learning rate, compensated by "infinitely many" steps, so we get a differential equation for the parameters, and a learning "velocity" $\alpha(t)$. Then we get as differential equation

$$\frac{\partial}{\partial t} b_i(t) = -\frac{\alpha(t)}{1 + e^{f(\vec{x}_0)}} \cdot u_i \cdot \sin(d_i + b_i(t))$$

$$\frac{\partial}{\partial t} u_i(t) = \frac{\alpha(t)}{1 + e^{f(\vec{x}_0)}} \cdot \cos(d_i + b_i(t)) \tag{2}$$

We assume we can train as long as we want, and we are not interested in how fast this procedure is proceeding. This means we are only interested in the direction of the vectors in (equation 2) and can as well look at the re-scaled vectors giving the differential equation

$$\frac{\partial}{\partial t} b_i(t) = -u_i \cdot \sin(d_i + b_i(t))$$

$$\frac{\partial}{\partial t} u_i(t) = \cos(d_i + b_i(t)) \tag{3}$$

These equations are independent for each neuron $i$, we we can fix a neuron $i$ and drop the subscript. Introducing the function

$$c(t) := \cos(d + b(t))$$

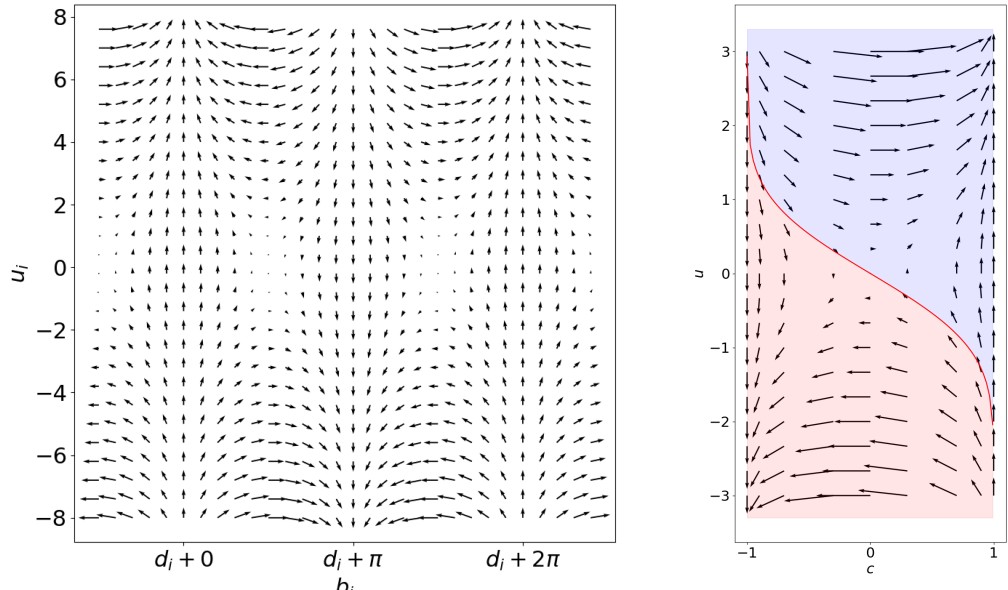

Figure 7: Vector fields for differential equations (equation 3), (equation 4)

and using the dot notation for the time derivative we get the simple differential equation

$$\begin{aligned} \dot{c} &= u \cdot (1 - c^2) \\ \dot{u} &= c \end{aligned} \tag{4}$$

(See figure 7 for a plot of the corresponding vector fields.)

**Lemma 1:** For (Lebesgue–)almost all $(c_0, u_0) \in [-1, 1] \times \mathbb{R}$ the solution curve of (equation 4) starting at $t = 0$ at $(c_0, u_0)$ will go either to $(1, +\infty)$ or to $(-1, -\infty)$ for $t \to \infty$.

**Proof of Lemma 1:**

Since we have a symmetry $(c, u) \mapsto (-c, -u)$, it is enough to consider $u_0 \geq 0$.

For $c_0 \in \{\pm 1\}$ we obviously have $c$ constant and $u(t) = u_0 + t \cdot c_0 \to \pm\infty$.

So we will assume $|c| < 1$. For $0 < c_0 < 1, 0 < u_0$ the $c(t)$ is increasing, hence we must have $u(t) \geq u_0 + t \cdot c_0 \to +\infty$. So we can assume also $u_t \geq 1$. Then $c(t)$ must be above the solution to the differential equation

$$\dot{f} = 1 - f^2,$$

which has the solutions

$$f(t) = \frac{e^{2t} - a}{e^{2t} + a},$$

so we also must have $1 \geq c(t) \geq f(t) \to 1$, hence $c(t) \to 1$.

For $c_0 = 0, u_0 > 0$ or $u_0 = 0, c_0 > 0$ we move for small $t > 0$ into the domain $c > 0, u > 0$, so we know that for them as well we have $c(t) \to 1, u(t) \to +\infty$. By symmetry, for $u_0 = 0, c_0 < 0$ we will have $c(t) \to -1, u(t) \to -\infty$.

For $c_0 = 0, u_0 = 0$ the functions $c, u$ are constant 0, so this is a point we have to exclude.

So the remaining part is $-1 < c_0 < 0, 0 < u_0$. From (equation 4) we see that in that case $\dot{c} > 0$ and $\dot{u} < 0$. So a solution curve starting in the quadrant $-1 < c_0 < 0, 0 < u_0$ has to leave this quadrant by either

1. Going through a point $c = 0, u > 0$, (light blue in figure 7), or
2. Going through a point $c < 0, u = 0$, (light red in figure 7), or
3. approaching $c = 0, u = 0$ (red line in figure 7).

We have already seen that the first two possibilities lead to $c(t), u(t) \to 1, +\infty$ and $c(t), u(t) \to -1, -\infty$ respectively, so it remains to show that the points that will approach $c = 0, u = 0$ are a null set. In fact, we show that for each $-1 < c_0 < 0$ there can be at most one $u_0$ with that property: From (equation 4) we see that the tangent to a solution curve through a point $(c_0, u_0)$ has slope $u \cdot c/(1 - c)$. If we look at the vertical distance between two solution curves in our quadrant, this means that with increasing $c$ their distance also increases. Therefore, two curves with positive distance at $c_0$ cannot converge both to $(0, 0)$, and for each given $c_0$ there can be at most one $u_0$ in our quadrant such that $(c_0, u_0)$ belongs to the third set above. ∎

With Lemma 1 we can now prove:

**Proposition 1**: Assume the weight vectors $\vec{w}_i$ are sampled from a normal distribution $\mathcal{N}(0, \Sigma)$ and then fixed, the numbers $b_i$ and $u_i$ are sampled independently from some distribution (not identically zero) with finite second moments (e.g. a normal distribution) and then trained on one data point $\vec{x}_0$ with label 1, then Gradient Descent (for the usual cross-entropy loss) will make the network's output converge to a function that is approximately proportional to $e^{-\vec{x}^T \Sigma \vec{x}/2}$. This approximation becomes exact when the number of neurons approaches infinity, or when we take the expected value.

**Proof:**

We assume that we have trained equation 2 for some time $T$ long enough such that we are in the situation of Lemma 1, by symmetry (changing $u_i \mapsto -u_i$, $b_i \mapsto b_i + \pi$ if necessary) we can assume that we have always $u_i \gg 0$ and $\cos(d_i + b_i) \approx 1$. Setting $Z := \sum_i u_i$, we have

$$\frac{1}{Z} f(\vec{x}) \quad \approx \quad \frac{1}{Z} \sum_{i=1}^{N} u_i \cdot \cos\left(\vec{w}_i \cdot (\vec{x} - \vec{x}_0)\right) \tag{5}$$

$$\approx \quad \mathbb{E}_{\vec{w} \sim \mathcal{N}(0, \Sigma)}\left[\cos\left(\vec{w} \cdot (\vec{x} - \vec{x}_0)\right)\right] \tag{6}$$

(For the second approximation we use that we have sampled the $\vec{w}_i$ independent from the $u_i$.) This is then the Fourier transform of the Gauss function and we get

$$\frac{1}{Z} f(\vec{x}) \approx e^{-\vec{x}^T \Sigma \vec{x}/2}.$$

The approximation (equation 5) becomes exact when we train infinitely long, and (equation 6) when we have infinitely many neurons (or take the expected value). ∎

**Proposition 2**: Under the same idealized conditions (infinite number of neurons, weights $\vec{w}_i$ frozen), but arbitrary finite training set, and assuming the training converged, the function goes to 0 for $|\vec{x}| \to \infty$.

**Proof:** We assume we trained on $M$ points $x_1, ..., x_M$ (with the correct label) and use a shallow Fourier net with $N \approx \infty$ neurons. We also assume that the training "converged" in the sense that the $b_j$ converged (the $u_j$ keep growing). Analog to the above case of one training point, we have

$$\frac{\partial L}{\partial b_j} = -\sum_{k=1}^{M} \frac{1}{1 + e^{f(\vec{x}_k)}} \cdot u_j \cdot \sin(\vec{w}_j \cdot \vec{x}_k + b_j).$$

With

$$e_k := \frac{1}{1 + e^{f(\vec{x}_k)}}, \qquad E(\vec{w}) := \sum_{k=1}^{M} e_k \cdot e^{i\vec{w} \cdot \vec{x}_k}$$

we can write this using the imaginary part $\Im$:

$$\frac{\partial L}{\partial b_j} = -u_j \Im\left(e^{ib_j} E(\vec{w}_j)\right).$$

If $E(\vec{w}_j) = 0$, the value of $L$ does not depend on $b_j$; we first show that this only happens for a (Lebesque–)Null set of $\vec{w}$.

The real part $\Re(E)$ is a real analytic function, and its value at 0 does not vanish since

$$E(0) = \sum_{k=1}^{M} e_k > 0.$$

Therefore the zero set of $\Re(E)$ has Lebesgue measure zero (an exercise in Calculus II, according to, and solved in (Mityagin, 2015)). This implies that the even smaller zero set $\mathcal{Z} := \{\vec{w} \in \mathbb{R}^d \mid E(\vec{w}) = 0\}$ also has Lebesgue measure zero.

Using the arg function

$$\arg : \mathbb{C} \setminus \{0\} \to \mathbb{R}/2\pi\mathbb{Z}, \; r \cdot e^{i\phi} \mapsto \phi$$

we can define the continuous function

$$B : \mathbb{R}^d \setminus \mathcal{Z} \to \mathbb{R}/2\pi\mathbb{Z}, \; \vec{w} \mapsto \arg(E(\vec{w}))$$

which makes

$$e^{iB} : \mathbb{R}^d \setminus \mathcal{Z} \to \mathbb{C}, \; \vec{w} \mapsto e^{i \cdot B(\vec{w})}$$

a well defined continuous function.

Now for $\vec{w} \in \mathbb{R}^d \setminus \mathcal{Z}$ and $u_j \neq 0$

$$\frac{\partial L}{\partial b_j} = -u_j \Im \left( e^{ib_j} E(\vec{w}_j) \right)$$

depends on $b_j \in \mathbb{R}/2\pi$, and $L$ is minimal if this derivative crosses from negative to positive values around $b_j$, which happens exactly when

$$b_j = \begin{cases} -B(\vec{w}_j) & u_j > 0 \\ -B(\vec{w}_j) + \pi & u_j < 0 \end{cases}$$

So for almost all $\vec{w}_j$ the weight $\vec{w}_j$ determines the phase $b_j$ in a way that depends continuously on $\vec{w}_j$. With this our function becomes

$$\sum_j |u_j| \cdot \cos\left(\vec{w} \cdot \vec{x} - B(\vec{w}_j)\right)$$

and after rescaling by dividing by $Z := \sum_i |u_i|$ this converges in the limit of infinitely many neurons to

$$f(\vec{x}) := \mathbb{E}_{\vec{w} \sim \mathcal{N}(0,\Sigma)} \left[ \cos(\vec{w} \cdot \vec{x} - B(\vec{w})) \right].$$

With the Gauss function

$$\phi_\Sigma(\vec{w}) := \frac{e^{-\vec{x}^T \Sigma \vec{x}}}{\sqrt{(2\pi)^d \cdot \det(\Sigma)}}$$

giving the probability density function for $\mathcal{N}(0, \Sigma)$ this is

$$\begin{aligned}
f(\vec{x}) &= \int_{\mathbb{R}^d} \phi_\Sigma(\vec{w}) \cdot \cos(\vec{w} \cdot \vec{x} - B(\vec{w})) \, d\vec{w} \\
&= \Re\left( \int_{\mathbb{R}^d} \phi_\Sigma(\vec{w}) \cdot e^{i\vec{w}\cdot\vec{x}} \cdot e^{-iB(\vec{w})} \, d\vec{w} \right) \\
&= \Re\left( \int_{\mathbb{R}^d} \phi_\Sigma(\vec{w}) \cdot e^{-iB(\vec{w})} \cdot e^{i\vec{w}\cdot\vec{x}} \, d\vec{w} \right),
\end{aligned}$$

which is up to a multiplicative constant the real part of the Fourier transform of the function

$$g(\vec{w}) := \phi_\Sigma(\vec{w}) \cdot e^{-iB(\vec{w})}.$$

Since $|e^{-iB(\vec{w})}| = 1$ and $\phi_\Sigma \in L^1(\mathbb{R}^d)$, this function is also in $L^1(\mathbb{R}^d)$, so the Riemann–Lebesgue Lemma gives that $f(\vec{x}) \to 0$ for $|\vec{x}| \to \infty$ as claimed. $\blacksquare$

## D RELU NETWORKS IN THE SIMPLEST 1-D PROBLEM

After randomly initializing the weight $w_i \sim \mathcal{N}(0, \sigma)$ and bias $b_i \sim \mathcal{N}(0, \sigma)$ for neuron $i$, the location of the resulting kink of $ReLU(w_i \cdot x + b_i)$ is at $-b_i/w_i$, so it is Cauchy-distributed with center 0 and part of them will fall on the right hand side of $x_0$, and another part on the left hand side. Training the parameters tries to increase the value at $x_0$, but it will not move the kink across $x_0$. So we will with high probability (probability 1 for infinitely wide networks) see the sum of functions of which some increase for large positive $x$, others increase for very negative $X$, so in general the expected picture is indeed curving upward on both sides, like the experimental result in the first image of figure 2.

## E FEW POINTS

We construct as small set of points around -5, -2, 3, 9:

$$S = \{-5.3, -5., -5.1, \ -2.1, -2., -1.88, \ 3.2, 3.1, \ 8.95, \ 9.1, 9.22\}$$

The Fourier approximation gives figure 8.

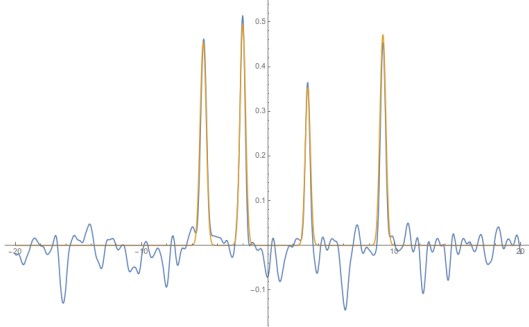

Figure 8: Fourier approximation for set $S$ of four small clusters in 1D

## F GAUSS NETWORKS

We can define "Gauss networks" for which we take as activation functions Gauss functions

$$g(x) := e^{-x^2/2}$$

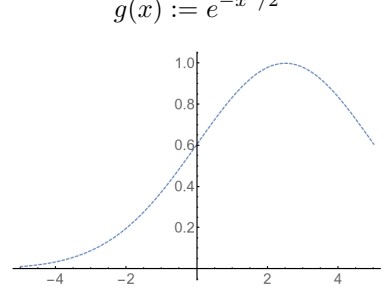

Figure 9: Example function $g(0.4 \cdot x - 1)$

Using this function instead of $ReLU(x)$ or $\sin(x)$ gives in the 2-dim toy example the figure 10 (compare with figure 4).

## G COMPARISON TO NEAREST NEIGHBOR METHODS

The "nearest neighbor" classifier assigns a label $L$ to a test input if the closest training input had label $L$. We can also use the "nearest neighbor" method for out of distribution detection: For a

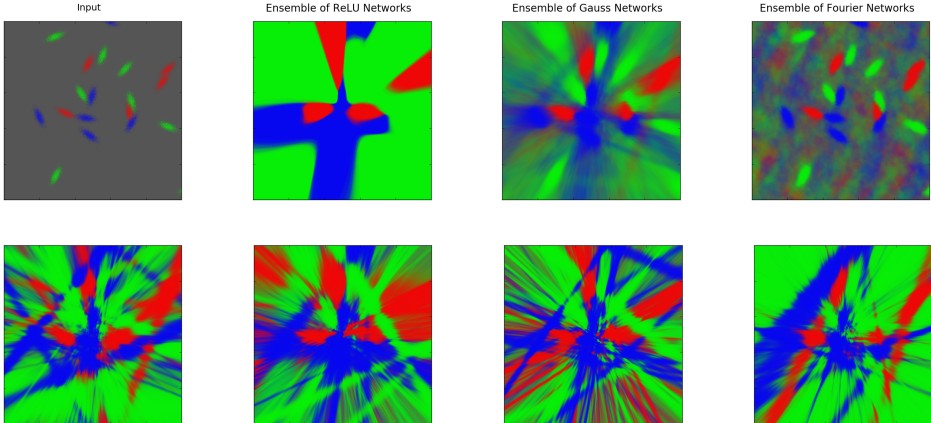

Figure 10: Upper row: Input, Average output for ensemble of 50 ReLU / Gauss / Fourier networks. Lower row: Predictions of 4 individual randomly initialized Gauss networks.

threshold $r$ we call an input "in distribution" if and only if its nearest neighbor in the training set has distance $\leq r$. Varying the $r$ then gives a ROC curve, analogous to varying the threshold $\theta$ for the softmax output of a network ensemble.

There are many variations of this simple method ($k$–nearest neighbor, RBF networks, adjusting the metric to the data), but for the general principles we want to discuss here the differences do not matter and we will use the basic implementation.

In (Bishop, 1995), chapter 5.9, pp. 183–186 the author writes about the "major potential difficulties with radial basis function networks" stemming from the "localized nature" of the hidden units (in our implementation of the nearest neighbor method the hidden units would correspond to the training points).

The problem is that nearest neighbor methods "just memorize" training points, and only generalize to nearby points – this requires that we have a training point "near" any possible input point of our training distribution. On the other hand, "normal" networks learn "rules", which can generalize the "important parts" and ignore the "noise" that does not contribute to the classification.

This can be a problem for nearest neighbor methods mainly in high dimensions, but as a first illustration we reproduce the two–dimensional toy problem from the book: Assume we have three classes (red, green, blue) that can be distinguished by their $x_1$–coordinate, but we also have a random $x_2$–coordinate with high variance (see figure 11).

If the red / green / blue stripes have length $\delta$ along the $x_1$–axis, a nearest neighbor classifier or RBF network may pick circles of radius around $\delta/2$ to characterize the points that belong to the same stripe. However, for that we need to have enough points also in $x_2$ direction, if there is a gap of much more than $\delta$, we will have points in a stripe that are not recognized as belonging to the training distribution (or to the corresponding label).

On the other hand, a ReLU network will learn some regions and (over)confidently classify all points in these regions as red / green / blue.

We can demonstrate in this toy example how the Fourier network "interpolates" between a "ReLU behavior" and a "Nearest Neighbor behavior" when we adjust the standard deviation used in the weights' initialization of a Fourier network: For small initializations, we are close to the "generalizing" ReLU behavior, large initializations give something close to a "memorizing" Nearest Neighbor behavior. We show in figure 12 how an ensemble of ReLU networks and ensembles of Fourier Networks generalize to the 2D plane. The upper row gives the classification around the input, the lower row zooms out, in particular showing that for Fourier networks the ensembles are not extending their confidence region to infinity (as the ReLU networks do).

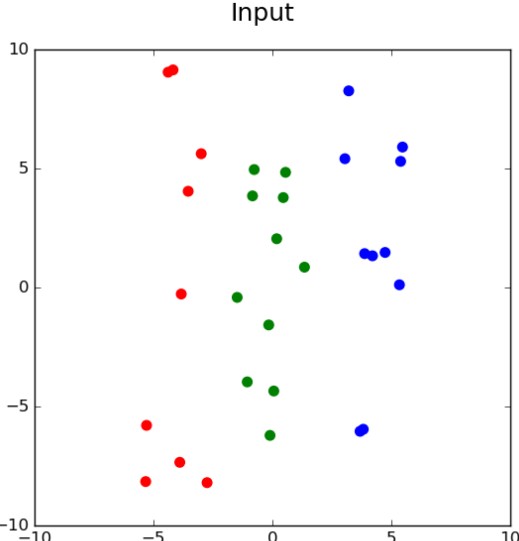

Figure 11: Input - the points on the left have label "red", in the middle "green", and on the right "blue".

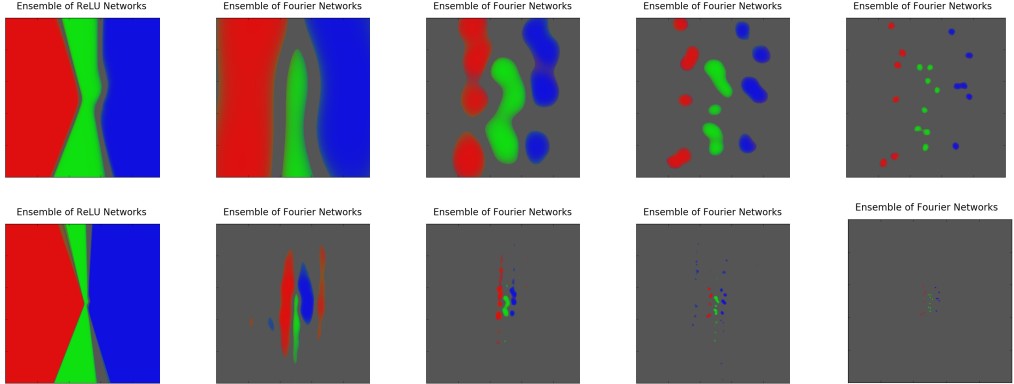

Figure 12: From left to right: Output of ReLU network, and Fourier networks with $\sigma = 0.15, 0.55, 1, 2$. Areas with confidence $< 0.75$ are grayed out.
Upper row: Range [-10, 10]; Lower row: Zoomed out to range [-50, 50]

To illustrate the resulting effect in higher dimensions, we add some "random background" to the MNIST and fashionMNIST images (see figure 13).

The "random background" plays the role of the $x_2$ coordinate above: For a new input, the nearest neighbor classifier now has to find a training image that is close to the new one both in the digit and in the background. This is of course much harder (even with 60000 training points), and correspondingly the classification accuracy of the nearest neighbor classifier drops from 96.9% to 76.2% (see table 4). For ReLU (and Fourier) nets, we also see a drop, but only to 91.3%, the classifier just learns to "ignore" the background. (The classification accuracies reported here are the accuracy of the averaged softmax output of the networks in an ensemble.)

For out of distribution detection, we see the same effect – for the nearest neighbors method the area under the ROC curve drops from 98.9% to 77%, but for ReLU nets only from 91.8% to 91.1%. The Fourier networks are supposed to combine the best of both approaches, and indeed they give better out–of–distribution detection than either method in both cases.

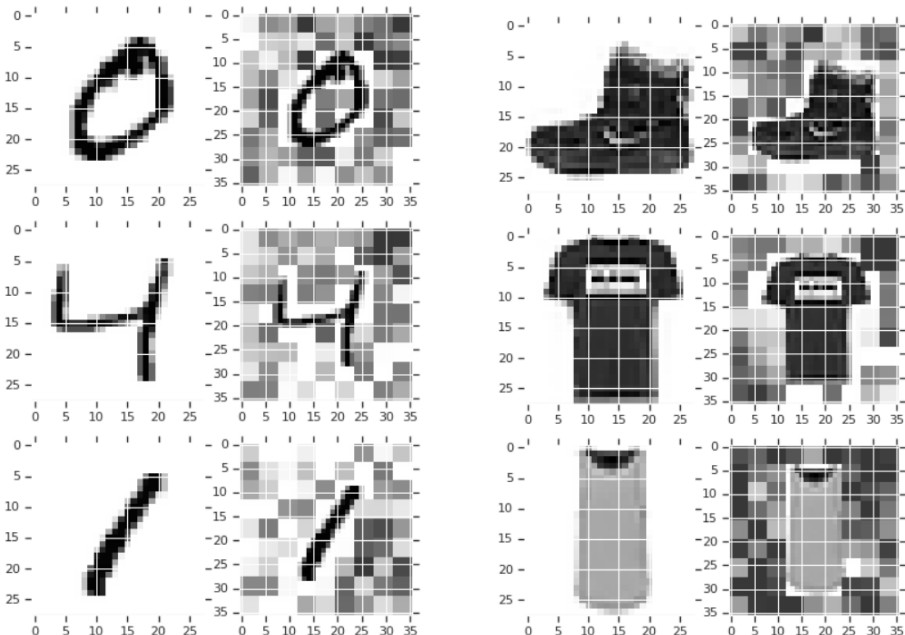

Figure 13: Embedding MNIST / fashionMNIST into a random background

| Metric | Area under ROC | | Classifier accuracy | |
|---|---|---|---|---|
| Model | Original | With background | Original | With background |
| ReLU nets | 91.8% | 91.1% | 98.2% | 91.3% |
| Fourier nets | **99.7%** | **95.8%** | 97.5% | 91.2% |
| Nearest Neighbor | 98.9% | 77.0% | 96.9% | 76.2% |

Table 4: Comparison between an ensemble of one-layer ReLU networks, an ensemble of one-layer Fourier networks and just using the distance to the nearest neighbor in feature space (which in this case coincides with the pixel space) as a score for OOD detection. The in-distribution dataset is MNIST while the out-of-distribution data is coming from FashionMNIST.

As a second example, we produce 60000 wider images by randomly selecting 4 MNIST digits and combining them to one new image. As the label of these images we take the label of the first digit. For out–of–sample images we replace the first digit by a fashionMNIST image, see figure 14.

Again, this makes it much harder for nearest neighbor method — now it has to find images in which all 4 digits are similar (which is still possible with 60000 training images, but harder), whereas the ReLU classifier can focus on the pixels in the first quarter of the image and learn to ignore the rest, see table 5 for the results.

An interesting philosophical question arises when we produce images by exchanging the last digit by a fashionMNIST image: Are the resulting images "outliers" or not? In any case, the results in table 5 show that the ReLU classifiers now do not flag many "outliers", which is in agreement with our expectation that they focus on the left quarter of the image and mostly ignore the rest.

| Metric | Area under ROC | | Classifier accuracy |
|---|---|---|---|
| Model | First digit fashion | (last digit fashion) | Original |
| ReLU nets | 85.4% | (59.0%) | 94% |
| Fourier nets | **95.2%** | (79.1%) | 92% |
| Nearest Neighbor | 79.4% | (79.7%) | 57% |

Table 5: As in table 4, but for the "4 digits" data set.

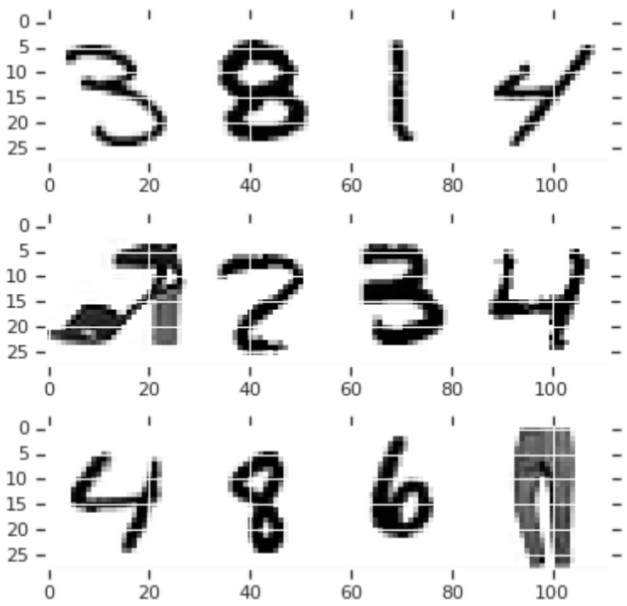

Figure 14: 4 digits

# H   ReLU NETWORKS WITH INFINITESIMAL INITIALIZATION

For given training data and infinitesimal initialization, there are only finitely many possibilities to which networks can converge (independent of network size), see (Maennel et al., 2018). The arguments given in that paper show also that with increasing number of weights we should with high probability end up at the same possibility.

# I   MORE RESULTS ON MNIST

## I.1   VARIATION OF ROC CURVES FOR MNIST VS NOTMNIST AND FASHIONMNIST

The ROC curves obtained when considering MNIST as in-distribution and notMNIST samples as out-of-distribution and using two-layer classifiers for training were shown in figure 1. To visualize the variation of these curves, we compute 50 networks, and apply this method 10 times on ensembles of 10 randomly selected networks, this gives the left hand graph in figure 15. (The right hand graph in figure 15 shows the corresponding results for testing for "outliers" from fashionMNIST. We also get similar figures when training on fashionMNIST, see appendix M.)

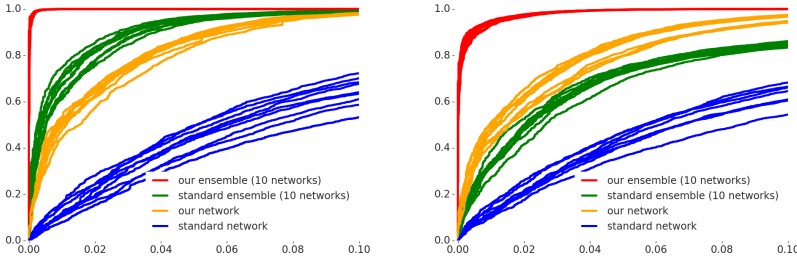

Figure 15: 10 random samples of ROC curves for networks ensembles trained on MNIST,
  evaluated on notMNIST (left) and fashionMNIST (right).
  $x$-axis: MNIST samples not recognized as belonging to MNIST,
  $y$-axis: notMNIST samples recognized as not belonging to MNIST.

## I.2 Entropy histograms

To compare our method also to the evaluation given in (Lakshminarayanan et al., 2017), we compute for each image the entropy of the softmax output and plot a histogram over the MNIST and the notMNIST data set for the ensemble sizes 1,5,10. The first three columns of figure 16 (copied from figure 3 in (Lakshminarayanan et al., 2017)) show the resulting entropy of the softmax output for ReLU networks, first using the ensemble as is, then with additional adversarial training, and in the third column the corresponding results for "dropout at test time".

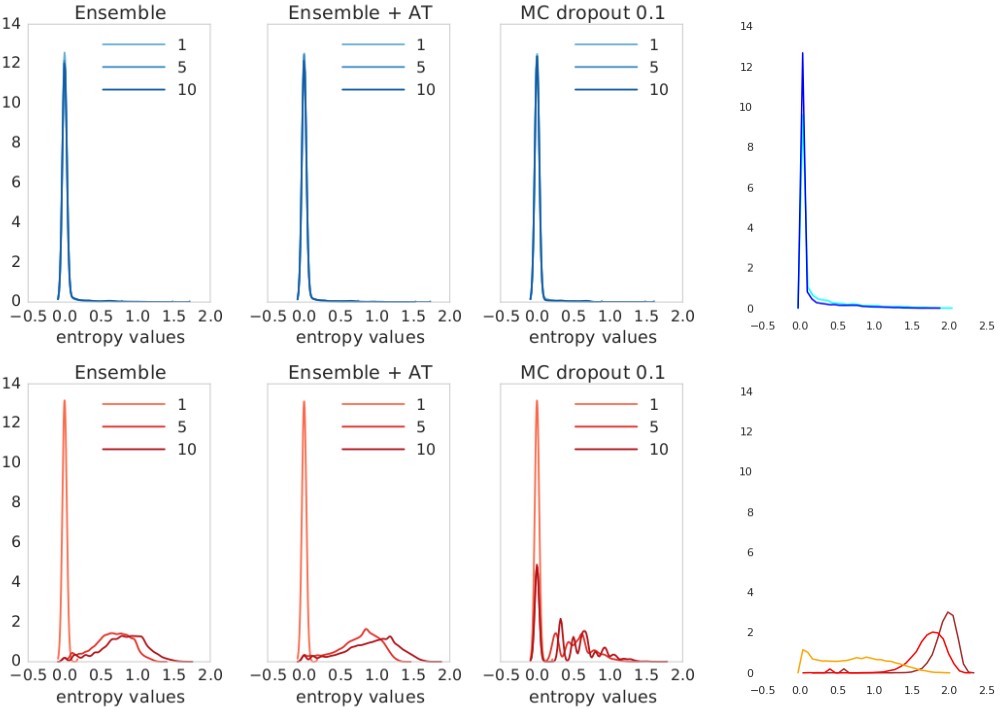

Figure 16: Histogram of the predictive entropy from figure 3 in (Lakshminarayanan et al., 2017) (first three columns) compared to the corresponding Fourier networks (last column). Top row: Known classes (MNIST), bottom row: Unknown classes (notMNIST).

For input from the MNIST test set (first row, blue curves) the entropy is almost always close to 0, i.e. the classifier puts almost all probability on one label. For input from the notMNIST data set the classifier ensembles distribute the probability over several labels, resulting in much higher entropy. Use of adversarial training (second column) increases the entropy even a bit further for the notMNIST examples.

The fourth column shows the results for the corresponding Fourier networks. On the MNIST test set the behavior is basically the same as the usual network, but on notMNIST even one network often is unsure about the label, and an ensemble of 10 networks gives most of the time an entropy around 2 (the maximum would be $\log(10) \approx 2.3$ for the uniform distribution across all 10 labels).

## I.3 Convolutional networks on MNIST

Since MNIST vs. notMNIST is too easy to profit from better classifiers, we use a more difficult problem: Train on classes 0-4, see whether we detect classes 5-9 as "out of distribution".

For MNIST we use two convolutional layers with $7 \times 7$ windows and max pooling in $3 \times 3$ windows with stride 2, and add two fully connected layers with 200 / 100 neurons. We also use data augmentation by randomly cropping to $24 \times 24$ pixels. For the Fourier network we change the activation function of the last layer to $\sin(x)$.

First experiments seem to indicate that a good way to choose the initialization for the different layers of the Fourier network is to start with small initializations for the first layers and the last layer, and then choose the second last initialization as large as possible (see appendix J.5 for details).

With this setup we get for the area under the ROC curve:

| Network | standard ReLU | ReLU, large init | $\sin(x)$ |
|---|---|---|---|
| convolutional | 95.9% | 98.1% | 99.1% |
| fully connected | 95.2% | 96.2 | 96.3% |

Table 6: AUROC on MNIST 0-4 vs MNIST 5-9.

# J DETAILED SPECIFICATIONS OF THE NEURAL NETWORKS AND THEIR PARAMETERS

## J.1 ONE DIMENSIONAL TOY EXAMPLE

- Input data:
  Only one point at $x_0 = 2$ with label "1".
- Network:
  One hidden layer with 500 neurons, weights initialized with $\sigma = 0.0632 \approx \sqrt{2/500}$.
- Learning:
  - ReLU: 500 steps with learning rate 0.001
  - sin: 5000 steps with learning rate 0.001

## J.2 TWO DIMENSIONAL TOY EXAMPLE

- Input data:
  20 centers randomly chosen in $-10 \leq x \leq 10$, $-10 \leq y \leq 10$. To each center one of the labels "Red", "Green", "Blue" was randomly chosen. For each center $(x, y)$ there were 50000 points randomly drawn from a normal distribution $\mathcal{N}(\mu, \Sigma)$, where $\Sigma = R \cdot D^2$ for a randomly drawn rotation matrix $R$ and $D$ the diagonal matrix with entries $\sigma_1 = 0.2$ and $\sigma_2 = 0.5$.
- Network:
  One hidden layer with 500 neurons, activation function sin or ReLU.
  Weights (including bias terms) initialized with $\sigma = 12$.
- Learning:
  26 epochs, batch size 256, learning rate 0.02.

## J.3 784 DIM INPUT, ONE HIDDEN LAYER (FULLY CONNECTED), TRAINED ON MNIST

Input: 764 pixels, normalized to be in $[-1, 1]$

Hidden layers: One hidden layer of 100 neurons

Initialization: $\sigma \in \{0.002, 0.01, 0.05, 0.075, 0.1, 0.125, 0.15, 0.175, 0.2, 0.22, 0.25,$
$$0.275, 0.3, 0.35, 0.4, 0.5, 1.0, 2.0, 4.0, 6.0\}$$

Batch size: 256

Learning rate: 200 epochs with 0.02, 200 epochs with 0.005.

## J.4 784 DIM INPUT, TWO HIDDEN LAYERS (FULLY CONNECTED), TRAINED ON MNIST

Input: 764 pixels, normalized to be in $[-1, 1]$

Hidden layers: Two hidden layer of 200 neurons, first layer: ReLU, second layer: ReLU / sin.

Initialization:

- For ReLU network: $\sigma_1 = 0.07, \sigma_2 = 0.14$.
- For Sine network: $\sigma_1 = 0.75, \sigma_2 = 0.0002$

Parameters searched: For ReLU this is the standard initialization $\sigma = 2/\sqrt{(n)}$.

For Sine networks, initial tests gave the result that for a given $\sigma_2$ the $\sigma_1$ should be as large as possible such that the training still converges. Then I tried $\sigma_2 \in \{0.0002, 0.001, 0.01\}$ and $\sigma_1 \in \{0.25, 0.5, 0.75, 1.0\}$. Learning rate was 0.01.

Best results: ($\sigma_1, \sigma_2$, area under ROC curve for MNIST vs. notMNIST):

0.5, 0.0002, 0.99898

0.5, 0.001, 0.99908

0.5, 0.01, 0.99946

0.75, 0.01, 0.99974

0.75, 0.001, 0.99977

0.75, 0.0002, 0.99977

## J.5  784 DIM INPUT, CONVOLUTIONAL NETWORKS, TRAINED ON MNIST

Input: 764 pixels, normalized to be in $[-1, 1]$,

Preprocessing: 4 times random cropping from $28 \times 28$ to $24 \times 24$.

Hidden layers:

- Convolution, $7 \times 7$ windows, $1 \rightarrow 16$ channels (to size $24 \times 24 \times 16$)
- Max pooling in $3 \times 3$ windows with stride 2 (to size $12 \times 12 \times 16$)
- Convolution, $5 \times 5$ windows, $16 \rightarrow 32$ channels (to size $12 \times 12 \times 32$)
- Max pooling in $3 \times 3$ windows with stride 2 (to size $6 \times 6 \times 32$)
- Fully connected, 200 neurons, ReLU activation function
- Fully connected, 100 neurons, ReLU or $\sin(x)$ activation function

Batch size: 40

Learning rate: 25 epochs 0.002, 20 epochs 0.001, 20 epochs 0.0005

Initialization: Given by $\sigma = (\sigma_1, \sigma_2, \sigma_3, \sigma_4)$.

For Fourier networks:

After initial attempts systematic evaluation of area under ROC curve for MNIST 0-4 vs. MNIST 5-9 for

- $\sigma = (0.001, 0.001, \sigma_3, 0.0001)$, best result: 99.1% for $\sigma_3 = 20$
- $\sigma = (0.002, 0.001, \sigma_3, 0.5)$, best result: 99.1% for $\sigma_3 = 1.5$
- $\sigma = (0.1, 0.1, \sigma_3, 0.001)$, best result: 98.8% for $\sigma_3 = 1.75$.

For ReLU networks:

After initial attempts systematic evaluation of area under ROC curve for MNIST 0-4 vs. MNIST 5-9 for

- $\sigma = (0.01, 0.01, \sigma_3, 1.0)$, best result: 97.8% for $\sigma_3 = 0.3$
- $\sigma = (0.01, 0.01, \sigma_3, 0.6)$, best result: 97.4% for $\sigma_3 = 0.3$
- $\sigma = (\sigma_1, \sigma_1, 0.1, 0.1)$, best result: 97.7% for $\sigma_1 = 0.9$
- $\sigma = (\sigma_1, \sigma_1, \sigma_1, \sigma_1)$, best result: 98.1% for $\sigma_1 = 0.55$

### J.6  SIMPLE CNN, TRAINED ON SVHN

Input: 32x32x3-shaped images, with the pixels normalized to be in $[0, 1]$,

Hidden layers:

- Convolution, $7 \times 7$ windows, $1 \to 16$ channels (to size $32 \times 32 \times 16$)
- Max pooling in $3 \times 3$ windows with stride 2 (to size $16 \times 16 \times 16$)
- Convolution, $5 \times 5$ windows, $16 \to 32$ channels (to size $16 \times 16 \times 32$)
- Max pooling in $3 \times 3$ windows with stride 2 (to size $8 \times 8 \times 32$)
- Fully connected, 200 neurons, ReLU activation function
- Fully connected, 200 or 100000 neurons, ReLU or $\sin(x)$ activation function

Batch size: 128

Learning rate: 0.01 (for 200 neurons on the last layer) and 0.001 (for 100000 neurons). Halved every 40 epochs.

Best $\sigma$ for initializing the last layer: 0.75.

### J.7  SHALLOW VGG, TRAINED ON SVHN

Input: 32x32x3-shaped images, with the pixels normalized to be in $[0, 1]$,

Hidden layers:

- the first 4 convolutional blocks of the VGG16 architecture (including the corresponding max pooling layers) (to size $8 \times 8 \times 128$)
- Fully connected, 200 or 100000 neurons, ReLU or $\sin(x)$ activation function

Batch size: 128

Learning rate: 0.001. Halved every 40 epochs.

Best $\sigma$ for initializing the last layer: 0.01.

### J.8  VGG16, TRAINED ON SVHN

Input: 32x32x3-shaped images, with the pixels normalized to be in $[0, 1]$,

Hidden layers:

- all the convolutional blocks of the original VGG16 architecture (to size $1 \times 1 \times 512$)
- Fully connected, 200 neurons, ReLU activation function
- Fully connected, 100 or 100000 neurons, ReLU or $\sin(x)$ activation function

Batch size: 128

Learning rate: 0.001. Halved every 40 epochs.

Best $\sigma$ for initializing the last layer: 0.25 and 0.1 for $sin(x)$ and ReLU respectively, for 100 neurons on the last layer and 0.002 for 100000 neurons.

## K    DEFINITION OF THE DATA SETS

As input distributions that are "clearly different" from MNIST we can use the fashionMNIST ((Xiao et al., 2017)) and notMnist ((Bulatov, 2011)) data sets, see figure 17.

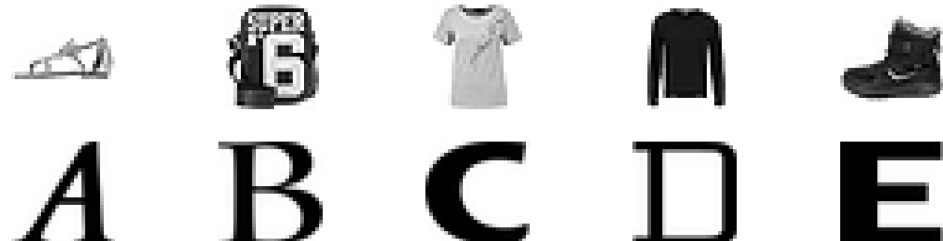

Figure 17: FashionMNIST (top) and notMNIST (bottom) data sets.

We also use some constructed data sets:

- "Masked fashionMNIST":
  In MNIST some pixels are always or almost always white, see figure 19. Out of fashion-MNIST we derive this data set by setting all pixels to 0 (white) that have an average gray value of below 0.01.

- "Masked notMNIST":
  This is derived from notMNIST in the same way.

- Circles and lines:
  Two lines and two circles in random positions.

- Flipped MNIST:
  Derived from MNIST by mirroring the digits 3,4,6,7,9 horizontally and the digits 4,6,7,9 vertically.

- IID noise:
  Uniform iid pixel values from [0.256].

- Independent pixels:
  Each pixel is randomly sampled from pixels at the same position at MNIST training images, independent of the other pixel values.

- PseudoMNIST: Here we try to mimic not only the value distribution for each pixel, but also local correlations. We start with an image in which each pixel is independently sampled like in the previous distribution. Then we perform 1000 (PseudoMNIST1) or 1500 (PseudoMNIST2) updates of the following form: We randomly select a $3 \times 3$ window, and find the closest match in 1000 randomly sampled images from MNIST, where "closest" is with respect to the L2 distance of the 8 outer pixels of the $3 \times 3$ window. Then we replace the middle pixel in our image with the value of the middle pixel in this "closest match" from MNIST.

We will also use a "masked version" with a mask from MNIST: see figure 19 for the origin of the mask, and figure 18 for examples.

## L    TRAINING ON MNIST, USING OTHER OUTLIER SETS

With these data sets we can again compute the ROC curves that show how well we can distinguish the training set from the outlier set by setting a threshold on the "confidence". As confidence we take again the averaged softmax output of the predicted label over an ensemble of 10 classifiers, in which the weights are randomly sampled from a normal distribution with mean 0 and standard

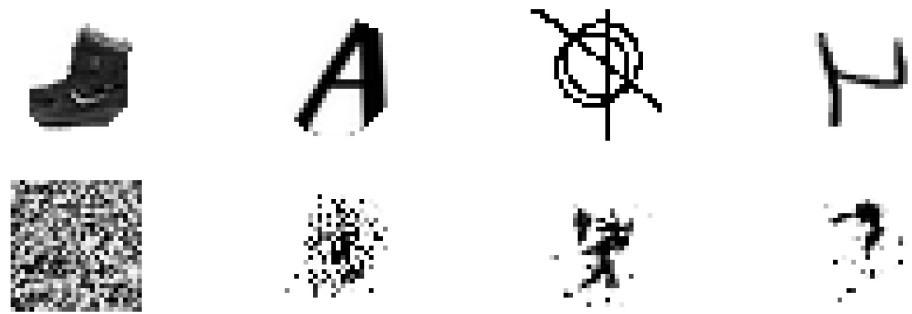

Figure 18: Constructed data sets.
Top row: Masked fashionMNIST, masked notMNIST, circles and lines, flipped MNIST.
Bottom row: IID noise, independent pixels, pseudoMNIST1, pseudoMNIST2.

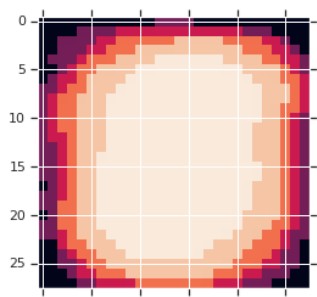

Figure 19: Average pixel value $\leq$ thresholds 0, 0.0001, 0.001, 0.01, 0.1 on MNIST

deviation $\sigma$. We consider the area above the ROC curve as "error" and plot $-\log(error)$ against the initialization $\sigma$ used for the networks. The resulting graphs are in figure 20.

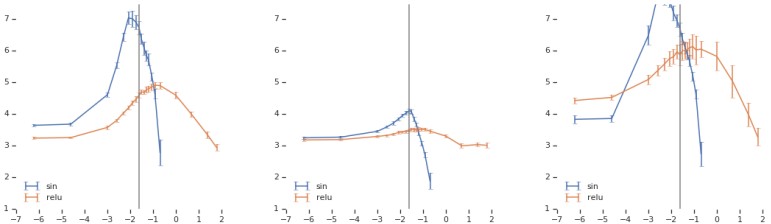

Figure 20: Influence of initialization
$x$-axis: $\log(\sigma)$, $y$-axis: $-\log(error)$, higher is better
Test sets: Circles and lines, pseudo mnist, IID noise.

In some cases, it is easy to distinguish MNIST samples from outliers by just looking at pixels close to the corners: In MNIST these pixels are always white, but in notMNIST often not. To eliminate this shortcut, we also consider "masked" versions of notMNIST and fashionMNIST in which these pixels are set to white (see appendix K for details). In figure 21 we show the corresponding graphs for the masked versions, and for images in which each pixel is independently sampled from MNIST images, so each pixel has the same value distribution as in MNIST. Even with the "looking at the corners shortcut" disabled, the Fourier networks still perform well.

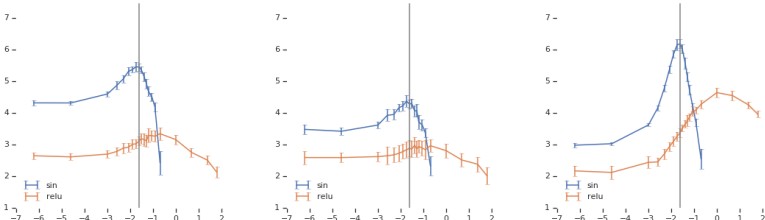

Figure 21: Influence of initialization, axes as in figure 5
Left to right: masked notMnist / fashion MNIST, pixels sampled independently from MNIST

## M  TRAINING ON FASHION MNIST

We use the same two hidden layer network architecture, initializations, and learning rates as in figure 1, but train on fashion MNIST, the result is in figure 22 below. Since this is a more challenging data set, the same architecture gives not as good results as for MNIST (note the $x$-axis extends now to 0.2), but the difference between ReLU and $\sin(x)$ networks is comparable, resulting in graphs (figure 22) that look very similar to figure 1.

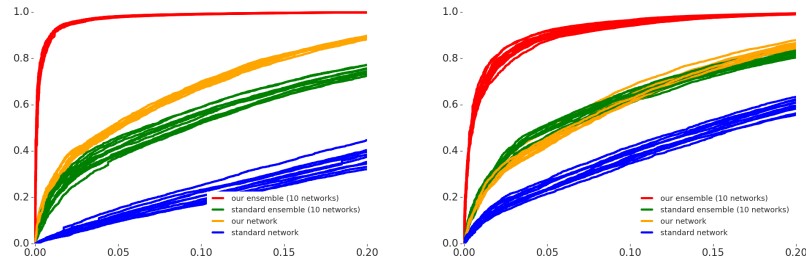

Figure 22: 10 random samples of ROC curves for networks ensembles trained on fashion MNIST, evaluated on notMNIST (left) and MNIST (right).
$x$-axis: MNIST samples not recognized as belonging to MNIST,
$y$-axis: notMNIST samples recognized as not belonging to MNIST.

We can also plot the area under the ROC curve for different initializations of a network with one hidden layer - this gives figure 23. Again this is qualitatively the same result as in figure 5 or in figure20 above.

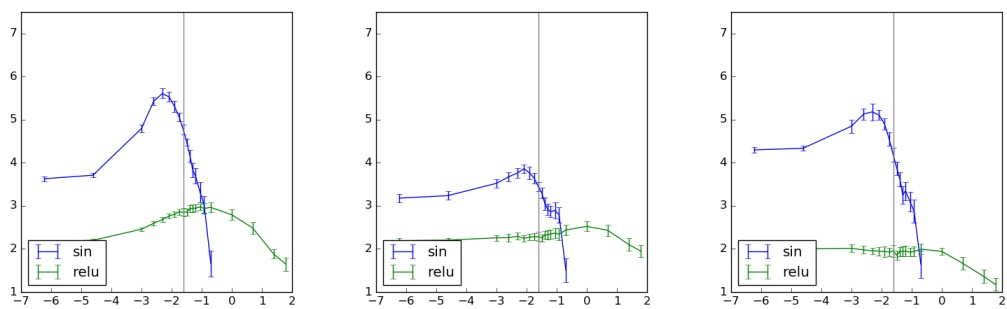

Figure 23: Influence of initialization, networks trained on fashion MNIST
$x$-axis: $\log(\sigma)$, $y$-axis: $-\log(error)$, higher is better
Test sets: Circles and lines, MNIST, notMnist.

# N    INFLUENCE OF THE DISTRIBUTION FROM WHICH WEIGHTS ARE SAMPLED

Our method for getting a variety of networks relies on different randomly sampled initializations giving different networks. According to (Sirignano & Spiliopoulos, 2018) this actually no longer happens if the number of neurons is very large: Then the networks converge to an idealized network in which there are infinitely many neurons that have their weights distributed according to the probability distribution from which we sample.

To illustrate this potential problem visually, we look at a very simple 2-dimensional toy example, in which we can again plot how the classifiers generalize from the training distribution to other points, like we did in section 3.

As input we use only a line, with labeled training samples given as blue dots in the middle and red dots at the end (figure 24).

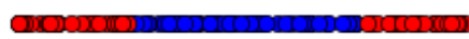

Figure 24: Input for the 2-dim example.

We will use a ReLU network with one hidden layer of 100 neurons to classify all points in the surrounding area of the plane. Colors encode the "probabilities" that the softmax computes from the output layer, with blue / red signifying "probability 0/1 of being blue". The result is in figure 25.



Figure 25: ReLU networks, extrapolating from a line to 2d.
Left: Output of three example networks, right: average softmax output over 50 networks
As in section 3, we see that most points belong with high confidence to on of the labels (here they are either blue or red), and there is only limited variation for classifiers initialized from different randomly chosen weights.

If we change the activation function of the hidden layer from to ReLU to $\sin(x)$, we obtain different results outside of the training distribution (figure 26). While individual points are still mostly red or blue, i.e. the individual classifiers are still "overconfident", the different classifiers now tend to disagree and averaging the softmax outputs over several of them we get much more realistic confidences for the Fourier networks than for the ReLU networks: Now the ensemble is only confident about the label for points in the vicinity of the training inputs.

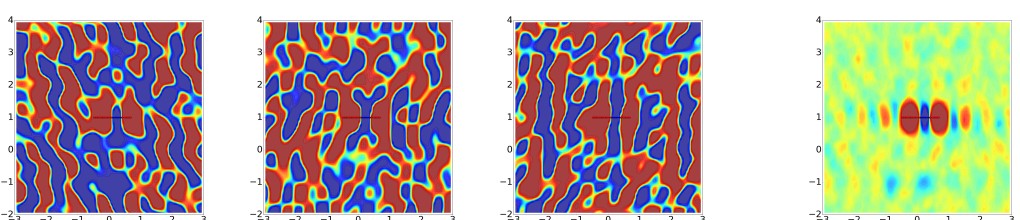

Figure 26: Fourier networks, extrapolating from a line to 2d.
Left: Output of three example networks, right: average softmax output over 50 networks

To demonstrate the potential problem with very wide networks, we now use 20000 neurons (instead of the 100 we used in the above pictures). Now the results of different initializations become more similar to each other, see figure 27.

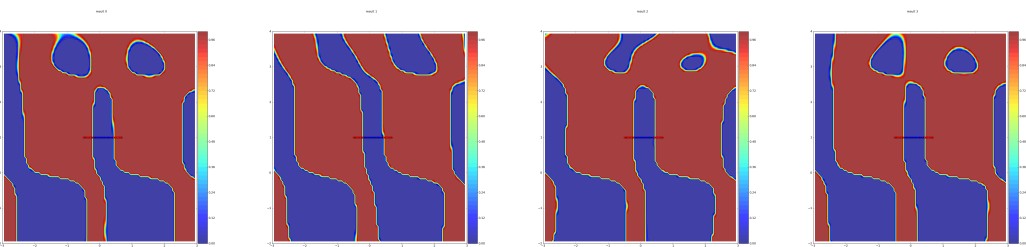

Figure 27: Different Fourier networks with weights initialized by samples from same distribution, 20000 hidden sin-neurons

So in this case, we can no longer rely on different sampling from the same distribution, but need to change the distribution itself. For higher dimensional input, this effect may be more theoretical because the number of neurons required to get into that regime is no longer realistic.

However, even for a moderate number of neurons it may make sense to vary the distribution of initial weights from which we sample to obtain a greater diversity. Here is an example in our case of 2-dimensional input, 100 neurons, ReLU activation:

The weights in the 2-dimensional case have 3 coordinates: Two for the direction and one for the bias. We can now sample the "direction" weights according to a Normal Distribution with one standard deviation $\sigma_1$, and the "bias" weight according to a Normal Distribution with a different standard deviation $\sigma_2$. Different pairs $(\sigma_1, \sigma_2)$ describe different distributions from which we initialize the weights.

If we vary now our weight initialization by varying $(\sigma_1, \sigma_2)$, the resulting networks seem to converge to different classifiers, see figure 28.



Figure 28: Ensemble average, 100 hidden ReLU-neurons
Left to right: $\sigma_1, \sigma_2 = (3, 1), (0.01, 0.01), (5, 5) (0.01, 1)$

In particular, the networks initialized by $\sigma_1, \sigma_2 = (5, 5)$ could in theory contain the same initializations as the networks initialized by $\sigma_1, \sigma_2 = (0.01, 1)$, but the emphasis shifts to different networks: The latter give blue cones extending downwards most of the time, whereas such a result is not among the 50 samples I obtained for $\sigma_1, \sigma_2 = (5, 5)$.

So even for not extremely large networks it would make sense to restrict the distribution of weights from which we sample in different ways to obtain a more diverse set of networks.

## O  SEPARATING DIFFERENT FEATURES FOR THE SAME LABEL

We try restricting the weights in a neural network classifier to certain subspaces – not only during initialization, but throughout the training. The hope would again be that classifiers with different restrictions develop different features which generalize in different ways to unseen examples, giving more variety for out-of-distribution samples.

In the MNIST case, we evaluate two strategies of restricting the weights of the first layer:

- Only allow access to half of the pixels.
  If each classifier sees only half of the image, they usually still can figure out which digit it is:

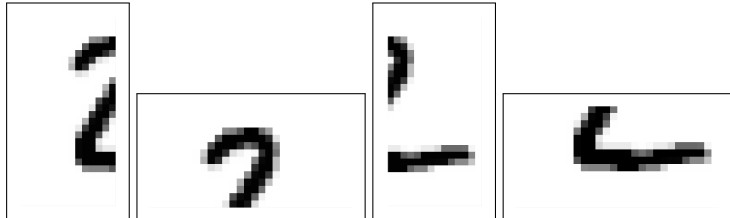

Figure 29: Restricting MNIST input to half of the image

- Only allow access to "line detectors" with certain directions.
  We implement this by convolution with a $7 \times 7$–matrix which corresponds to a stripe of a certain direction. We allow 4 (of 10) directions per location, and to reduce the dimension of the resulting weight space we subsample the locations to a grid, see figure 30. This again gives a vector space of dimension which is roughly half of the original dimension.

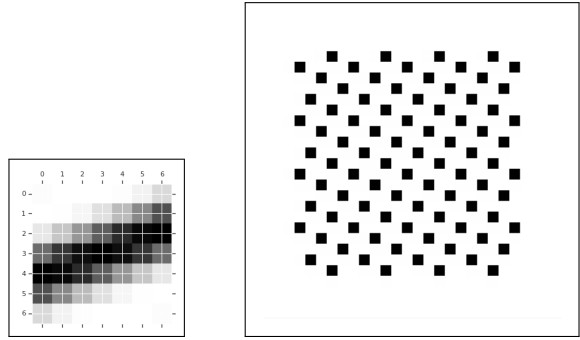

Figure 30: Convolution with stripes in some directions, centered at some locations

This seems to give a significant increase of the area under the ROC curve in particular for the difficult case "trained on 0-4, evaluate 5-9 as outliers":

Area under ROC curve

| "Outliers" | sin | sin with half input | sin with stripes | combined |
|---|---|---|---|---|
| MNIST 5-9 | 96.5% | 97.3% | 97.5% | 97.6% |
| fashionMNIST | 99.3% | 99.5% | 99.6% | 99.7% |
| notMNIST | 99.9% | 99.9% | 99.9% | 99.9% |

Same test with restriction to half images on fashionMNIST:

The classifier is trained on fashionMNIST 0-4.

| "Outliers" | relu, large init | relu, restr. | sin | sin, restr. |
|---|---|---|---|---|
| fashionMNIST 5-9 | 91.3% | 92.5% | 93.5% | 94.3% |
| MNIST | 94.2% | 95.1% | 98.9% | 99.2% |
| notMNIST | 93.4% | 95.2% | 99.6% | 99.6% |

We have also applied the same approach when training convolutional networks on SVHN. The table below shows the area under ROC curve obtained with a Simple CNN and a Shallow VGG. By applying a square mask on one of the corners of the image (top left, top right, bottom left, bottom right) we obtain an ensemble of constrained classifiers that is slightly more robust to out-of-distribution samples.

| Model | Original input | Masked input |
|---|---|---|
| Simple CNN | 96.6% | 96.5% |
| Shallow VGG | 95.8% | 96.6% |

## P    INPUT TO ACTIVATION FUNCTION

In this network the first hidden layer has ReLU neurons, and the second hidden layer has sin-Neurons. We plot the activation observed as input to the second layer, for the 5 neurons which have the largest weight going to the output of "0" (left), and the for the 5 neurons with the smallest weight going to the output of "0" (right) for all images that are indeed a "0". We observe that the peaks are separated by $2\pi$, the left at the maxima of sin, the right at the minima of sin. A similar pattern arises for other images from the training set.

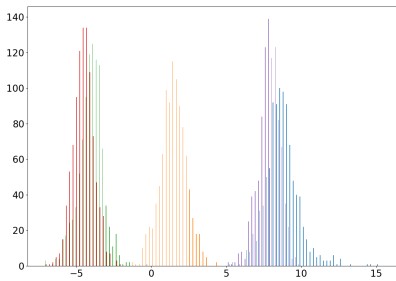 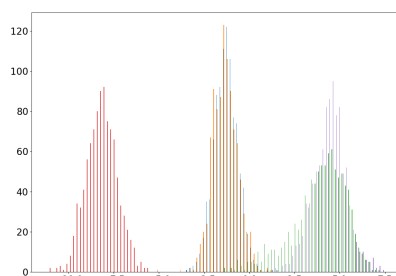

Figure 31: Activation function input for images of "0", left: largest , right: smallest weight.

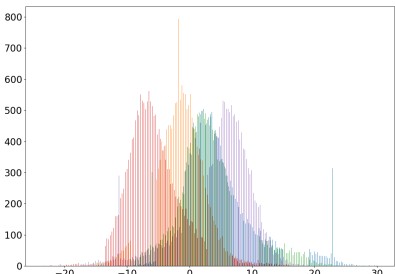

Figure 32: notMnist activation

However, when we plot the input to the sin activation for images from notMnist, we no longer see the peaks at maxima / minima of sin, suggesting that the output for these images will be "random":

## Q    MORE EXPERIMENTAL RESULTS FOR SVHN VS CIFAR10

Figure 33 shows which are the regions of optimal initialization for the final fully-connected layer for both ReLU and Fourier networks. Note that if the initialization is too large, gradient descent on the Simple VGG Fourier network will sometimes fail to converge.

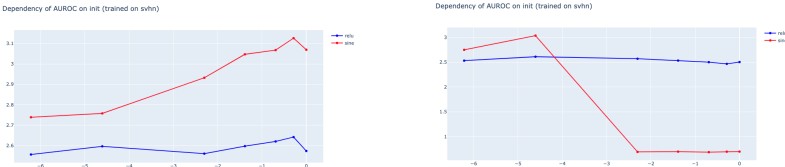

Figure 33: Influence of initialization on AUROC for SVHN, axes like in figure 5.

The distribution of the entropy of the softmax outputs over in-distribution and out-of-distribution samples for a Simple CNN network is shown in Figure 34. Results are presented for both regular ReLU and $sin(x)$ activations for the Simple CNN architectures (with $\sigma = 0.75$ used as the standard deviation of the initialization distribution of the last layer). The curves correspond to ensembles of 1, 5, and 10 models.

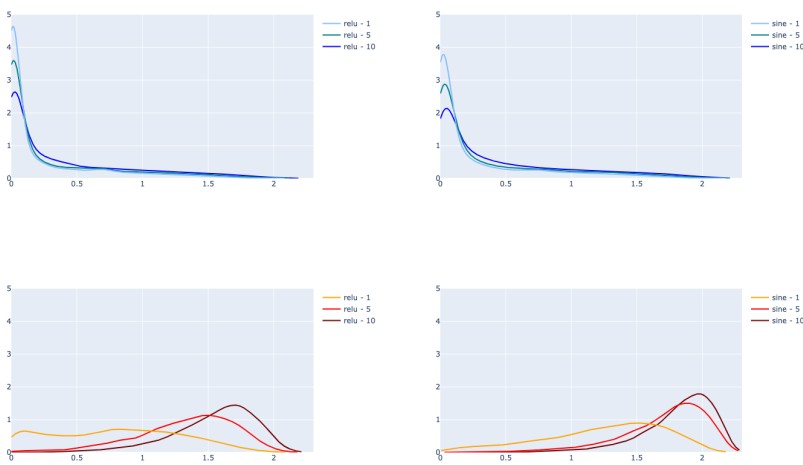

Figure 34: Entropy histograms for SVHN as in-distribution set (top row) and CIFAR10 as out-of-distribution set (bottom row). Comparison between using $sin(x)$ (right-hand side) or ReLU activations (left-hand side) on the last layer of a Simple CNN network.

Table 7 presents the best results obtained using convolutional networks trained on SVHN for both the ReLU and Fourier setup. Out-of-distribution samples are detected using two approaches, by selecting a threshold for either the maximum probability as obtained after aggregating the outputs of the softmax layers in an ensemble or for the entropy of the ensemble's prediction. For evaluation, we used the following metrics:

- the area under the ROC curve (*AUROC↑*)
- the false positive rate at 80% true positive rate (*FPR80↓*)
- the negative log-likelihood score (*NLL↓*)
- the Brier score (*Brier↓*)

In order to be closer to the ideal case, which is the premise for theoretical results proved above, we also report numbers for the same network architectures with significantly more neurons on the final fully-connected layer (i.e. 100,000). For each configuration, the table contains the numbers for the

model with the best value for the initialization $\sigma$. The dependency between the area under the ROC curve and the initialization strength is presented in figure 33.

| Network | Last Layer Activation | Last Layer Units | Last Layer Init $\sigma$ | AUROC↑ using max-p | FPR90↓ using max-p | AUROC↑ using entropy | FPR80↓ using entropy | NLL Score↓ | Brier Score↓ |
|---|---|---|---|---|---|---|---|---|---|
| Simple CNN | ReLU | 100000 | 1 | 91.97% | 12.89% | 92.18% | 11.82% | 0.3303 | 0.0136 |
| Simple CNN | sine | 100000 | 0.75 | **95.25%** | **7.80%** | **96.29%** | **5.55%** | 0.3324 | 0.0141 |
| Simple CNN | ReLU | 200 | 0.75 | 92.89% | 11.55% | 93.23% | 10.31% | 0.3058 | 0.0123 |
| Simple CNN | sine | 200 | 0.75 | **95.62%** | **7.38%** | **96.63%** | **5.18%** | 0.2895 | 0.0123 |
| Shallow VGG | ReLU | 100000 | 0.25 | 94.76% | 8.41% | 95.85% | 6.29% | 0.3512 | 0.0149 |
| Shallow VGG | sine | 100000 | 0.1 | **95.05%** | **8.34%** | **96.40%** | **5.67%** | 0.3094 | 0.0133 |
| Shallow VGG | ReLU | 200 | 0.01 | 92.65% | 12.21% | 93.51% | 9.97% | 0.3261 | 0.0138 |
| Shallow VGG | sine | 200 | 0.01 | **94.40%** | **9.56%** | **95.80%** | **6.72%** | 0.3184 | 0.0137 |
| VGG | ReLU | 100000 | 0.01 | 95.56% | 7.50% | **96.43%** | 5.31% | 0.2452 | 0.0091 |
| VGG | sine | 100000 | 0.002 | **95.69%** | **7.20%** | **96.49%** | **5.05%** | 0.2728 | 0.0090 |
| VGG | ReLU | 100 | 0.1 | 94.26% | 9.77% | 95.77% | 6.73% | 0.3348 | 0.0144 |
| VGG | sine | 100 | 0.25 | **94.63%** | **9.07%** | **95.88%** | **6.52%** | 0.3286 | 0.0143 |

Table 7: Extensive experimental results for ensembles of 5 models trained on SVHN and evaluated on CIFAR10 as out-of-distribution dataset.

## R    COMPARISON WITH OTHER OOD DETECTION METHODS

| Paper | Method | In-distrib. set | OOD set | AUROC |
|---|---|---|---|---|
| (Hendrycks & Gimpel, 2017) | baseline | MNIST | notMNIST | 93.2% |
| (Liang et al., 2018) | outlier exposure | MNIST | notMNIST | 98.2% |
| ours | Fourier networks | MNIST | notMNIST | **99.9%** |
| (Malinin & Gales, 2018) | outlier exposure | MNIST | FashionMNIST 5-9 | 96.5% |
| ours | Fourier networks | MNIST | FashionMNIST 5-9 | **99.7%** |

Table 8: Area under ROC curve for entropy as the OOD detection score.

(Hendrycks & Gimpel, 2017) use a slightly larger network and GELU activation functions, (Liang et al., 2018) train also on images that do not belong to MNIST, but also not on the "notMNIST" images that are used for evaluation. Similarly, (Malinin & Gales, 2018) use samples from a set of outliers to tune the model; the more similar these are to the outliers used for testing, the better the performance of the model is. Table 9 shows the results obtained using the Dirichlet Prior Network model (DPN) for various settings. In all instances we used the default hyperparameter values (suggested either in the paper or in the code released by the authors[1]). It is important to note that for the setting on which the Prior Network model performs best half of the out-of-distribution test samples are drawn from the outlier distribution that was used for fine-tuning the model (namely the distribution of FashionMNIST samples from classes 0-4, since we use the whole test set of FashionMNIST as OOD samples at test time). Moreover, for the best performing DPN results we tweaked the original setup and allowed the model to use the FashionMNIST training samples belonging to classes 0-4, instead of the (fewer) test samples, as prescribed by the authors of the paper. However, even on this significantly simplified setup, the DPN baseline falls behind the Fourier network ensemble even though the latter never sees any outliers during training.

| Model | In-distrib. | OOD - training | OOD - test | AUROC (entropy) | AUROC (max-prob) |
|---|---|---|---|---|---|
| DPN | MNIST | Omniglot | fMNIST 5-9 | 92.60% | 93.00% |
| DPN | MNIST | fMNIST 0-4 | fMNIST 5-9 | 96.50% | 96.60% |
| ReLU nets | MNIST | N/A | fMNIST 5-9 | 91.78% | 91.89% |
| Fourier nets | MNIST | N/A | fMNIST 5-9 | **99.69%** | **99.51%** |

Table 9: Comparison between ReLU / Fourier network ensembles and various settings of the Dirichlet Prior Network model.

---

[1]https://github.com/KaosEngineer/PriorNetworks-OLD

