# OpenReview forum: "Fourier networks for uncertainty estimates and out-of-distribution detection"
_ICLR.cc/2020/Conference — Reject_

### Official Review · AnonReviewer3 · 2019-10-23
**Official Blind Review #3**

**Rating:** 3

**Review:**

** Updates after rebuttal **

I thank the authors for the response, though I am still skeptical about the evaluation of the method, which might be a result of heavy tuning and overfit to the chosen test sets. The proposed approach also requires more theoretical justification.

------------------------------------------------

I'm not an expert in this area but I do find this paper interesting. Though the name "Fourier networks" is a bit arbitrary because the proposed approach also applies to multi-layer networks where only the last hidden layer has the proposed change.

The extrapolation problem of ReLU networks is an interesting point. I don't know previous works that point out this for out-of-distribution detection but it's worth figuring out if this observation has been made in the adversarial robustness community.

I do have several concerns, summarized below:
* On page 3 the fourier transform is motivated by that RBF networks "do not generalize as well as ReLU networks". I doubt if there is any evidence for this argument.
* I have some issue understanding proposition 1: what is w_i'? Only w_i is mentioned before.
* The "Fourier network" is not defined explicitly in the paper, which makes it hard to understand the architecture/algorithm details. If I understand it correctly, it is only about changing the activation function of the last hidden layer and large initialization, with everything else the same as the ReLU networks?
* How does the magic number "\sigma_1 = 0.75" and "\sigma_2 = 0.0002" come from? Did you search it by looking at the test performance? Is the performance sensitive w.r.t. the two parameters?

I'm willing to increase my score if the authors addressed my concerns.



**Experience Assessment:**

I do not know much about this area.

**Review Assessment: Checking Correctness Of Derivations And Theory:**

I assessed the sensibility of the derivations and theory.

**Review Assessment: Checking Correctness Of Experiments:**

I assessed the sensibility of the experiments.

**Review Assessment: Thoroughness In Paper Reading:**

I read the paper at least twice and used my best judgement in assessing the paper.

---

> ### Author Response · Authors · 2019-11-13
> **Response to Reviewer #3**
>
> * RBF networks:
> Thanks for flagging this, this is an important point that we should have explained (we have now added an appendix “G: Comparison to Nearest Neighbor methods” to discuss this in more detail).
> First, note we are only comparing the use of RBF / Fourier / ReLU networks in the last layer. Of course the generalization of the whole system depends also on other factors, including the architecture in the lower layers, but for this discussion we assume we have only one hidden layer that we vary.
> The basic problem with RBF networks or Nearest Neighbor methods is their “localized” nature, as explained in more detail e.g. in section 5.9 of Bishop’s book “Neural Networks for Pattern Recognition”: Since these methods “only memorize” known training points, we may need a lot of training points to cover the whole distribution, which can be a problem especially in high dimensions. On the other hand, ReLU networks learn “rules” which can generalize the “important parts” and ignore “noise”.
> To give a concrete example, we added (in the new appendix G) a random background to the MNIST and fashionMNIST pictures. This reduces the classification accuracy on MNIST for the (ReLU or Fourier) networks from 98% to 91%, but for the nearest neighbor method from 97% to 76% - the nearest neighbor method now has to find images similar in both the digit and the background, whereas the networks “only” have to learn to ignore the background. We see the same effect also in OOD detection: The area under the ROC curve is better for the nearest neighbor method in the case of the original images, but with the random background it becomes much worse than for the ReLU networks. In both cases the Fourier networks give better OOD detection than both ReLU networks and Nearest Neighbors.
> As another example we use images containing 4 digits, but the labels depend only on the first digit. To produce “outliers”, we exchange the first digit for a fashionMNIST image. Again the ReLU network is better in (classification and) OOD detection than the nearest neighbor method (0.85 vs. 0.79 area under ROC), and the Fourier Networks are better than both (0.95). The explanation “the ReLU networks learn to focus on the first digit” would lead us to predict that ReLU networks would not flag as many outliers if we instead changed the last digit to a fashionMNIST image. This is indeed the case. For details, see appendix G.
>
> * Proposition 1:
> Sorry, yes, that should be u_i (the weight connecting the sin(x) neurons to the output neurons), we corrected it in the new version.
>
> * “Fourier network”:
> Yes, with “Fourier network” we just mean using sin(x) as the activation function in the last layer. We added this as a definition.
>
> * Choice of initialization:
> The general recipe is to use the largest initialization that still gives a good accuracy of the classifier.
> For one hidden layer you can see the sensitivity with respect to the initialization in Figure 5 - the Fourier networks outperform the ReLU networks for any initialization as long as they still train reliably, and the gain seems to be highest the larger the initialization is (i.e. shortly before the performance drops because the networks do no longer train stably). So this setting can be found using only the training data, and Figures 5, 16, and 17 show that indeed the optimum does not differ significantly for different out of sample sets.
> For two layers we have two parameters we can tune, we can fix one and again choose the other as large as possible. So again you could use this recipe on the training set, appendix I.4 has empirical results which show that the results using this recipe with different choices for the fixed parameter are all very similar.

---

### Official Review · AnonReviewer1 · 2019-10-24
**Official Blind Review #1**

**Rating:** 6

**Review:**


I have read the reviews and the comments. Overall I am still positive about the paper and I have confirmed the rating.

======================

This paper proposes a method for the uncertainty estimates for Neural Network classifiers, specifically out-of-distribution detection. Previous methods use an ensemble of independently trained networks and average the softmax outputs. The authors investigate this method (ensembles of ReLU networks) and observe three fundamental limitations:
“Unreasonable” extrapolation, “unreasonable” agreement between the networks in an ensemble, and the filtering out of features that distinguish the training distribution from some out–of–distribution inputs, but do not contribute to the classification (CONSTANT FUNCTIONS ON THE TRAINING MANIFOLD).

To mitigate these problems the authors proposed the following:

- Changing the activation function of the last hidden layer to the sin(x) function, and they claimed that this is going to guard against overgeneralization.
- Use larger than usual initialization to increase the chances of obtaining more diverse networks for an ensemble.
- They claimed that this combines the out-of-distribution behavior from nearest neighbor methods with the generalization capabilities of neural networks, and achieves greatly improved out-of-distribution detection on standard data sets.

The paper addresses an important problem, out of distribution detection, by proposing a Fourier network which is somewhere between a ReLU network (small initialization) and a nearest neighbor classifier (large initialization). The authors claimed that this leads to an out-of-distribution detection which is better than either of them.

The paper is well written and easy to follow. The authors did an interesting and precise investigation in how to force the confidence score to decay like a Gauss function by proposing to use the Fourier transform of such a Gauss function. By doing so they get the advantage of ReLU (ability to generalize) and prevent the network to become arbitrarily certain of its classification for all points. However, the authors claimed that when they switch the activation function to sin(x) the increase of |x| will usually stop at the first maximum or minimum of sin which is (around \pi/2). However, the authors did not explain how they get this result (i.e the value \pi/2). It would be interesting if the authors could show results for the case greater than or less than (\pi/2) to show the difference.
In addition, Figure 3 shows that the ensemble of ReLU networks is overconfident in most of the area, whereas the ensemble of Fourier networks is only confident close to the input, and in the discussion of constant function of their training manifold the authors discuss some example.

I would like to ask the authors:

Did the Fourier networks learn the input distribution?
How are defined: usual initialization, small initialization and large initialization?

The experiment section is adequate. However, it would strengthen the paper if the authors compared against other approaches such as:
- Predictive uncertainty estimation via prior networks, NeurIPS 2018.
- Generative probabilistic novelty detection with adversarial autoencoders, NeurIPS 2018.


**Experience Assessment:**

I have published one or two papers in this area.

**Review Assessment: Checking Correctness Of Derivations And Theory:**

I assessed the sensibility of the derivations and theory.

**Review Assessment: Checking Correctness Of Experiments:**

I assessed the sensibility of the experiments.

**Review Assessment: Thoroughness In Paper Reading:**

I read the paper at least twice and used my best judgement in assessing the paper.

---

> ### Author Response · Authors · 2019-11-13
> **Response to Reviewer #1**
>
> - Did the Fourier networks learn the input distribution?
> Yes, in all the examples we looked at, they learned the labels as well as the ReLU networks (see also appendix B). There are small differences, sometimes one of ReLU(x) and sin(x) is slightly better than the other, but we did not observe any general trend.
>
> - The first maximum or minimum of sin:
> On the one hand this is an experimental result (see the new appendix P), but it is also the expected plausible dynamics of Gradient Descent: If a neuron has a positive / negative contribution to one label, its output should be increased / decreased when the feature is present at a sample with this label. This increase / decrease will continue until either the maximum / minimum is reached, or the feature or its connection to the output have changed significantly. So after the network reached a “stable state”, we expect that the sin(x) - Neurons that are used significantly in the output have reached a maximum / minimum of the sin(x) function.
>
> - How are defined: usual initialization, small initialization and large initialization?
> “Usual initialization”: “He initialization”: variance = 2 / fan_in.
> The “large initialization” that we use to evaluate our networks is in general the largest initialization such that the ensemble still gives a good accuracy (at some point the networks no longer train reliably and the accuracy and out-of-distribution detection goes down, but anything before that usually works, see e.g. figure 5).
> With “small initialization” we mean anything that behaves similar to the limit of “infinitesimal initialization”, this may still include the “usual initialization” (however, this term should only appear in qualitative statements that motivate our methods).
>
> - Predictive uncertainty estimation via prior networks, NeurIPS 2018.
> This method uses both in-distribution inputs and out-of-distribution inputs for training/tuning the model, whereas our method only uses in-distribution inputs.
> If one knows which out-of-distribution inputs to expect, using them is an easy option to greatly increase the performance. However, the performance of this approach depends on how well the “out of distribution training samples” match the “out of distribution test” samples.
> We evaluated the method described in the paper on distinguishing MNIST from fashionMNIST, classes 5-9.
> The main results for the area under the ROC curve are:
> ReLU nets: 91.8%
> DPN, trained on Omniglot as outliers: 92.6%
> DPN, trained on fashionMNIST classes 0-4: 96.5%
> Fourier nets:  99.7%
> We added a comparison between this method and ours in Table 1 as well as a more detailed discussion in Appendix R.
>
>
> - Generative probabilistic novelty detection with adversarial autoencoders, NeurIPS 2018.
> The approach in this paper (GPND) makes use of an interesting yet complex model that is more computationally demanding to train and evaluate. The approach employs two separate adversarial losses which makes the whole system much more delicate to train. Similarly to our method, the training procedure does not require out-of-distribution samples.
> It is important to note that all of the experiments in the paper (with one exception) have been set up such that training is performed only on one class of the dataset and the remaining classes are considered to be outliers. Since our model relies on classifiers, we cannot reproduce this setup with our method, since we need more than one class in the training set to obtain the models for an ensemble. With the GPND method, we obtained an area under the ROC curve of 98% when using one MNIST class as in-distribution data and an equally large set of images from the other classes as out-of-distribution data. However, this number is not directly comparable with the results obtained with our experimental setup on MNIST. Alternatively, we trained the GPND method such that several classes are considered as inliers e.g. classes 0-4 from MNIST, with the remaining classes being used at test time as outliers. We obtained 76% as the area under the ROC curve with this setup for GPND, which is far from the ~99% that we achieve with Fourier networks using the same in-distribution and out-of-distribution sets (see Table 5 in Appendix I.3).
> It may be possible to get the GPND model to work for in-distribution sets that contain more class clusters by some heavy hyperparameter tuning, or it may be necessary to make some fundamental changes like replacing the GAN used for reconstruction with a conditional GAN to model several different manifolds.

---

### Official Review · AnonReviewer2 · 2019-10-25
**Official Blind Review #2**

**Rating:** 1

**Review:**

The paper presents a method to detect out-of-distribution or anomalous data points. They argue that Fourier networks have lower confidence and thus better estimates of uncertainty in areas far away from training data. They also argue for using “large” initializations in the first layers and sin(x) as the activation function for the final hidden layer.

The paper does not seem to have any significant logical reasoning on why their specific architecture works, but "describes" what they did. It is not clear what the novelty is, besides that they found an architecture that seems to work. Additionally while Fourier networks have lower confidence, that does not necessarily mean they are more accurate estimates of uncertainty. However the reviewer does acknowledge that the estimates are mostly likely better than ReLU networks that are well known for having terrible estimates of uncertainty.


**Experience Assessment:**

I have published one or two papers in this area.

**Review Assessment: Checking Correctness Of Derivations And Theory:**

I assessed the sensibility of the derivations and theory.

**Review Assessment: Checking Correctness Of Experiments:**

I assessed the sensibility of the experiments.

**Review Assessment: Thoroughness In Paper Reading:**

I read the paper at least twice and used my best judgement in assessing the paper.

---

> ### Author Response · Authors · 2019-11-13
> **Response to Reviewer #2**
>
> - Logical reasoning vs. describing:
> We feel we do give a logical reasoning; it involves analyzing three deficiencies of ReLU network ensembles and deriving modifications to avoid them:
>
> Section 2 (“Unreasonable extrapolation”): Our motivation for the “Fourier networks” is the known fact that ReLU networks become more confident away from the training set. Instead we want the “evidence functions” (logits) for each label to decrease with distance to the training points. This can be achieved by a RBF network with Gauss functions, but to get better generalization we would prefer to achieve a similar behaviour with a “normal” network.
> Our basis to achieve this is that the Fourier transform of a Gauss function is again a Gauss function: Proposition 1 says we can get “evidence” functions that decay like a Gauss function around the point 0 also as the expected value of cos(wx) with w sampled according to the corresponding normal distribution, which in turn can be seen as the output of a network with cos(x) (or sin(x)) activation function, trained to have maximal output at 0. This leads us directly to networks with activation function sin(x) and weights that are sampled from a Normal Distribution.
> Proposition 2 then indicates that this indeed gives evidence functions that approach 0 away from the training points.
>
> Section 3: (“Unreasonable agreement between networks”): We sketch a mathematical argument why different “infinitesimal” initializations of ReLU networks with one hidden layer give the same network with probability 1, which would be bad for the ensemble approach. Avoiding this is one motivation for larger initialization.
> While “large initialization” and “sin(x) activation function” both invalidate essential parts of the mathematical argument, we rely on experiments to show that this “unreasonable agreement” is really gone with our modification.
>
> In section 4 we sketch a mathematical argument why features that do not contribute to the discrimination between labels are “dropped” when we start from “infinitesimal initialization”, and how these features do contribute to OOD detection when we start from “large initialization”.
>
> Admittedly, these pieces of mathematical reasoning do not completely “prove that it works” since we make some simplifying assumptions: Proposition 2 assumes we freeze the frequencies after sampling, which is a good approximation to the real procedure (Gradient Descent also on the frequencies) only for large initializations. Similarly, the proof of section 3 assumes “infinitesimal initializations” and only is valid for one hidden layer, and the argument of section 4 does not specify the magnitude of this effect. However, they motivate the approach and at least make it plausible that it would work. We feel this is the most useful level of mathematical rigor; it is rare in this area that one can rigorously prove general theorems without simplifying assumptions.
>
>
> - Novelty:
> We think the new contributions of this paper are:
> - An analysis of the limitations of the usual Ensemble approach for out-of-distribution detection.
> - The general idea of using the Fourier transform for mimicking a nearest neighbor method as an average of networks, thus combining “generalization” properties of networks with “boundedness” properties of nearest neighbor methods.
> - The use of this approach to significantly improve the performance of Ensemble methods without making them more complicated.
>
>
> - Lower confidence and accurate estimates of uncertainty:
> There seems to be a misunderstanding: There are two separate metrics one could use to measure the usefulness of “confidence scores”:
> “Information content” as measured by the ROC curve
> “Calibration”
> We use the first measure in our evaluation, it measures directly how well we can distinguish between “in distribution” and “out of distribution”, and it is invariant under monotonic transformations of the confidence scores used (and can even be used for e.g. nearest neighbor methods that do not have confidence scores that can be thought of as probabilities). So we do not evaluate a “lower confidence” (or only in the sense that out-of-distribution samples should get a lower confidence than in-distribution samples).
>
> Independent of this, one could also ask about the second evaluation - when the confidence scores can be interpreted as “probabilities”, it would measure how close these confidence scores are to the fraction of correctly classified in-distribution samples. For out-of-distribution samples, we do not have a correct label, and would instead require all labels to get the same probability 1/#labels.
> We are less interested in this evaluation since the calibration can always be adjusted later by a monotonic transformation and is more interesting for in-distribution misclassifications, so we did not include it in the paper.
> But for this evaluation, a low maximal softmax output is indeed what is desired and measured for out of distribution samples.

---

### Public Comment · ~Pranav_Poduval1 · 2019-09-26
**Interesting, but how is this even working ??**

Large Initializations will obviously add more diversity because of the highly non-convex NN optimization so that part is pretty obvious.

The other more important issue- Let's take an ensemble of NN with a single layer, so the last layer which is the first layer will have sin(x) as activation fn.
Now if all ur networks have learned to classify a particular training point x correctly, then I can create infinite adversaries that are out of training distribution as x+n*pi, n is any Integer.
I am sure as we go deeper it will not be so trivial to create adversaries, but it's clearly not that hard either.
Have I misunderstood something??
Possibly normalizing the last layer o/p between [-pi,pi] could be a soln. are you doing the same

---

> ### Author Response · Authors · 2019-09-26
> **Multiple frequencies**
>
> 1) In the simplest case 1-dim input x, 1 hidden layer with N neurons, the output of neuron j would be   sin(w_j*x+b_j) with the weights w_1,...,w_N between input and hidden layer (in general) different numbers.
> So when you choose a neuron j, the points   x+2*pi*n/w_j   for intergers n would give the same output for this neuron, but (in general) not for the other neurons (which would be needed to guarantee the same result as for x in the output layer).
>
> 2) You could still use the idea of your construction to create points that give approximately the same outputs because they are at most an epsilon away from x+2*pi*n/w_j for all j=1,2,...,N. However, their density decreases (in general) exponentially with N, as O(epsilon^N), for a single network with a single hidden layer of N neurons, so you would not encounter these points just by chance.
>
> 3) The aim of this paper is to reduce as far as possible the cases in which "random" out-of-distribution inputs are treated as in-distribution. It is not about defending against "carefully crafted" adversarial inputs. (In this case, "adversaries" are sort of the opposite of the usual "adversarial inputs": Instead of a point close to a training point x which gets a different output label than x, I assume you mean a point that is far away from the training set, but gets the same output as x).

---

> > ### Public Comment · ~Pranav_Poduval1 · 2019-09-27
> > **Indeed Adverserial Attack and Defence wasn't the point of the paper**
> >
> > The reason for my strong criticism was because most works focusing on uncertainty, also tend to show benefits of their methods in case of Adverial Robustness or detecting Adversarial Attacks e.g. Evidential Deep Learning,  Alpha-Divergence Dropouts etc.

---

> > > ### Author Response · Authors · 2019-09-27
> > > **Relationship to defending against Adversarial Attacks**
> > >
> > > As mentioned above in 3), there are two different meanings of adversarial:
> > > a) the one you used in your first question (far away point, same label), and
> > > b) the usual adversarial attack and defense (near point, different label).
> > >
> > > Indeed b) is sometimes (but not always) related to uncertainty and out-of-distribution detection.
> > > Actually, this work started with a defense against adversarial attacks, but it used a (both mathematically and computationally) more complicated method. We then noticed that it also can be used for out-of-distribution detection, and for out-of-distribution detection (but not for the defense against adversarial attacks) the method presented here performs as well as the more advanced method. So while there is a connection between the two problems also for this method, we think for this type of methods it may make sense to treat both goals differently.

---

### Decision · Program_Chairs · 2019-12-19

**Decision:**

Reject

**Comment:**

This paper presents a new method for detecting out-of-distribution (OOD) samples.

A reviewer pointed out that the paper discovers an interesting finding and the addressed problem is important. On the other hand, other reviewers pointed out theoretical/empirical justifications are limited.

In particular, I think that experimental supports why the proposed method is superior beyond the existing ones are limited. I encourages the authors to consider more scenarios of OOD detection (e.g., datasets and architectures) and more baselines as the problem of measuring the confidence of neural networks or detecting outliers have rich literature. This would guide more comprehensive understandings on the proposed method.

Hence, I recommend rejection.